# FOCUS ON LIKELY CLASSES FOR TEST-TIME PREDICTION

## ABSTRACT

We ask: Can focusing on likely classes of a single, in-domain sample improve model predictions? Prior work argued "no". We put forward a novel rationale in favor of "yes": Sharedness of features among classes indicates their reliability for a single sample. We aim for an affirmative answer without using hand-engineered augmentations or auxiliary tasks. We propose two novel test-time fine-tuning methods to improve uncertain model predictions. Instead of greedily selecting the most likely class, we introduce an additional step, *focus on the likely classes*, to refine predictions. By applying a single gradient descent step with a large learning rate, we refine predictions when an initial forward pass indicates high uncertainty. The experimental evaluation demonstrates accuracy gains for one of our methods on average, which emphasizes shared features among likely classes. The gains are confirmed across diverse text and image domain models.

## 1 INTRODUCTION

State-of-the-art optimization methods in supervised and self-supervised learning, prevalent across subdisciplines ranging from image recognition in computer vision to text generation with large language models, typically minimize cross-entropy loss. The optimization objective aligns with the ideal scenario where a model assigns probability one to the correct class and zero to all others, achieving perfect class discrimination. Such outcomes are achievable on training data since neural networks can memorize even random data (Zhang et al., 2021). However, during test-time, a model might exhibit (as we also show) uncertainty, particularly in cases where it errs. Nevertheless, current decoding strategies still greedily select the most likely class. For domain adaptation, related approaches to ours such as minimizing entropy and distinguishing certain and uncertain samples have not shown accuracy gains in general (e.g.,Wang et al. (2020); Hu et al. (2025b)). In fact, for our setup it was argued that *Focusing on likely classes [through entropy minimization] leads to a trivial solution confirming just the most likely classWang et al. (2020)*. The statement is reasonable as the optimization objective of (Wang et al., 2020) and all related works (including ours) is indeed to make the likely classes even more likely. This rationale paired with the obvious fact that absolute gains seem inherently very limited when optimizing just one single in-domain sample (before a model reset) probably kept researchers from pursuing this avenue and kept them focused on domain adaptation requiring auxiliary tasks or data (see Table 1). All of which are undesirable.In this work, we question the "wisdom" that optimizing towards a few likely classes leads to no gains through (i) a novel rationale, (ii) novel algorithmic variations (compared to domain adaptation) and (iii) extensive evaluation. Based on an in-domain mindset, we derive our key rationale: Whether a feature is shared or not between classes for a single sample impacts its reliability. Algorithmically, we propose that before making a choice in cases of high uncertainty, one should reflect on the estimated class distribution and narrow down the options by focusing on the most likely classes through fine-tuning of the network. We aim to contrast likely and unlikely outcomes through optimization, aiming to eliminate unlikely choices from consideration. The high-level idea is illustrated in Figure 1. Most domain-adaptation methods focus on minimizing entropy directly, while in the case where probability mass is heavily concentrated among the top-2 classes, which commonly holds, we preserve entropy (if we forgo weighting).

We introduce an *additional step focusing on likely classes during the prediction process if the initial forward pass yields high uncertainty*, as illustrated in the overview in Figure 2. We apply gradient descent at test-time in two distinct ways, either by *Decreasing outputs of Out-of-Focus (doFo)* towards

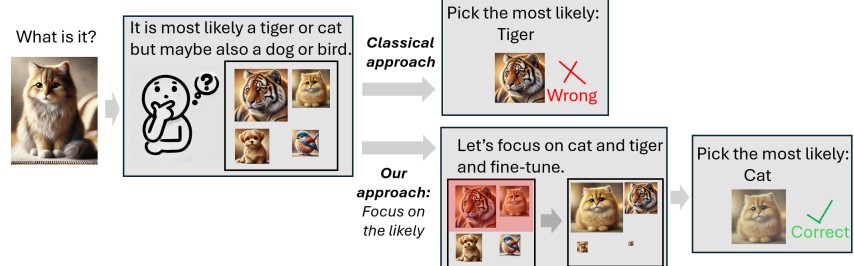

Figure 1: Overview: Fine-tune based on likely initial classes.

| Paper | In-Domain | | 1 Test Sample | | No extra needs |
| | Key topic | Eval | Per batch / Eval | Model Reset | |
|---|---|---|---|---|---|
| Cotta, TTT++, TSD+MLSC | ✗ | ✗ | ✗/✗ | ✗ | ✗ |
| TTT, ReCAP, DeYO | ✗ | ✗ | ✓/✓ | ✗ | ✗ |
| Memo | (✓) | (✓) | ✓/✗ | ✗ | ✗ |
| Tent, PASLE | ✗ | ✓ | ✓/✗ | ✗ | ✓ |
| SAR | ✗ | ✗ | ✓/✓ | ✗ | ✓ |
| **Ours** | ✓ | ✓ | ✓/✓ | ✓ | ✓ |

Table 1: Works on similar problems discussed in related work.

0 or by *Increasing outputs for Focus classes (iFo)*. While both methods aim at the same goal, i.e., focus on the likely, they follow different rationales, e.g., amplifying or lowering shared features among different sets of classes. It is not clear that these methods yield gains.

As the naive optimization method is computationally very expensive due to the need to conduct potentially tens or more of forward and backward passes per sample, we employ an uncertainty assessment step to limit our method to cases where it likely helps. Furthermore, we only perform a single extra forward and backward pass per sample, which yields similar outcomes as performing many iterations. We evaluate our methods across multiple datasets and classifiers on text generation and image recognition tasks. We demonstrate that iFo, which relies on enhancing shared features among likely classes, improves prediction accuracy in the majority of over 70 model-dataset pairs, whereas doFo, which suppresses features of unlikely classes, produces no gains.

## 2 METHODOLOGY

Next, we describe how we aim to improve predictions for high-uncertainty cases at test-time, as outlined in Figure 2. We add two components to the classical (single-forward) prediction process: (i) An *uncertainty assessment* procedure which decides whether to apply the focus optimization or not; (ii) A *focus optimization* module, which alters predictions.

**Uncertainty Assessment:** The prediction uncertainty assessment determines whether focus optimization should be applied to a specific sample or not. Focus optimization introduces overhead that

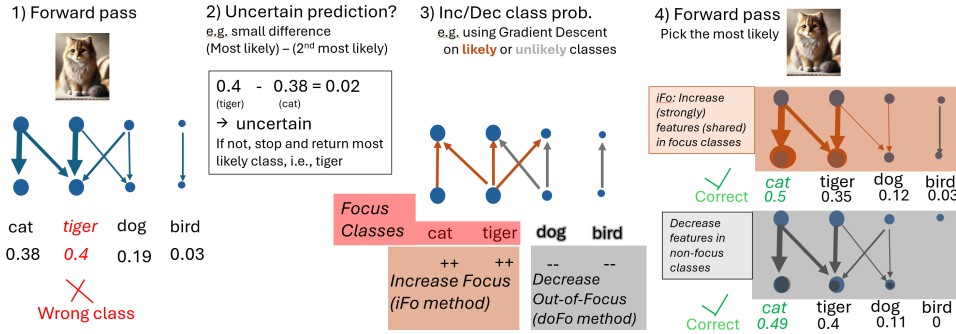

Figure 2: Our test-time approaches (iFo/doFo): If a prediction exhibits high uncertainty, changes are done to focus on likely (focus) classes either by increasing focus classes or decreasing all others.

should be avoided if it is unlikely to result in meaningful changes, particularly when uncertainty is low. We quantify uncertainty based on how strongly a classifier favors the predicted class compared to other classes. Specifically, we measure the uncertainty as the difference between the probabilities of the most likely and second-most likely classes. If this difference is large, the prediction is intuitively likely correct, as alternative classes have significantly lower probabilities. Concretely, we consider a model $f$, such as a neural network, that outputs logit values $f_c(X)$ for each class $c \in C$. The softmax layer converts the logit values $f_c(X)$ into probability estimates $p_c(X)$. Let $m(i)$ denote the class with the $i$-th highest probability for input $X$; thus, $p_{m(1)}(X)$ and $p_{m(2)}(X)$ represent the highest and second-highest probabilities, respectively. We define the difference as:

$$\Delta_{1,2} = p_{m(1)}(X) - p_{m(2)}(X) \tag{1}$$

When $\Delta_{1,2}$ is large, the model $f$ is quite certain that the most likely class is correct. Hence, focus optimization is unlikely to improve. Thus, we apply our optimization technique for a sample $X$ if $\Delta_{1,2} < d_{1,2}$ for a user-given threshold $d_{1,2}$. While softmax values can be poorly calibrated, we also care about what the network *assigns* rather than what really is as the chances of changing a class depend on the actual network outputs. Furthermore, softmax values incur almost no additional computation. Similar separations have been made for domain adaptation but with a performance goal rather than to save on computation (e.g., Hu et al. (2025a)).

**Focus Optimization:** We employ gradient descent for a fixed number of iterations, i.e., just one except for one experiment, to increase focus on the most likely classes, called "focus classes" $F \subset C$, where $C$ is the set of all classes. We do this for each sample anew on the original network as we require high learning rates that might negatively affect the network beyond the given sample. We discuss two seemingly opposing methods illustrated in Figure 2: i) *Decreasing outputs of Out-of-Focus classes $C \setminus F$ (doFo)* and ii) *Increasing outputs of Focus classes $F$ (iFo)*. While both methods share the same goal, they employ different rationales: iFo amplifies features shared among focus classes $F$, provided these features positively contribute to their likelihood. The underlying *assumption* is that if a feature positively contributes to the most likely predictions, it is likely highly relevant. This approach is motivated by the belief that features shared among multiple classes are generally more robust. In contrast, doFo suppresses features shared between focus and out-of-focus classes. The underlying *assumption* here is that features relevant to less likely classes are probably less important. This relies on the idea that these classes have accumulated significantly less probability mass, making their associated features potentially less credible. Visually, in the center panel of Figure 2, iFo increases activations of the subnetworks indicated by beige arrows, whereas doFo decreases those indicated by grey arrows. It is unclear whether either of these methods provides benefits. As we will demonstrate, improvements from doFo are inconsistent across tasks, while iFo performs more reliably. We set $|F| \geq 2$, meaning we focus on at least the two most likely classes for each sample $X$. Additionally, we apply gradient descent directly to the raw logits $f_c$ rather than to probabilities $p_c$, as indicated in Algorithm 1. When optimizing the log outputs $\log(p_c)$ from a softmax layer—typical in classical cross-entropy loss optimization—we must account for the normalization inherent in softmax. This results in a conceptual overlap of our approaches, simultaneously increasing focus-class probabilities and decreasing out-of-focus class probabilities, as discussed in the appendix. For iFo, we maximize the logits—and consequently the probabilities—of the focus classes $F$. For doFo, we minimize the logits of the out-of-focus classes $C \setminus F$.

*Weighting motivation:* A shortcoming of iFo is that increasing all focus classes in a naive manner increases unlikely classes relatively more than likely ones. In addition, it neglects our prior belief given by the initial forward pass yielding softmax probabilities preferring the most likely class. Thus, we add weighting by the softmax-probabilities $p_c$ for $iFo$. Without explicit weighting we would have an implicit weight of 1 for each focus class $F$ giving a total of $|F|$. We scale the probabilities $p_c$ used as weights by the number of focus classes $|F|$ to approximate the sum with implicit weighting, i.e., $\sum_{c \in F} |F| \cdot p_c \approx |F|$ as in most cases, most probability mass is in the focus classes. If not (i.e., $\sum_{c \in F} p_c \ll 1$), we are unsure that any of the focus classes is correct and we optimize much less towards them compared to no weighting. The weights $p_c$ are treated as constant in backpropagation denoted by $\tilde{p_c}$, meaning that gradients are not propagated through $p_c$.

Thus, we minimize the following losses:

$$L^{iFo}(X) = -\frac{\sum_{c \in F} |F| \cdot \tilde{p_c}(X) \cdot f_c(X)}{|F|} = -\sum_{c \in F} \tilde{p_c}(X) \cdot f_c(X), \quad L^{doFo}(x) = \frac{\sum_{c \in (C \setminus F)} f_c(X)}{|C \setminus F|} \tag{2}$$

These loss functions are among the simplest expressing our idea. The division to get a per-class average loss makes the learning rate less sensitive to the number of focus classes $|F|$. Due to the

---

**Algorithm 1** Focus on the Likely: iFo and doFo

---

**Require:** $f(\cdot|\theta)$: Model with parameters $\theta$;   $X$: Input;   $\eta$: learning rate;   Opt $\in \{\text{iFo}, \text{doFo}\}$;
   $T$: Iterations (default: 1); $n_f$: # focus classes (def.: 2);   $d_{1,2}$: uncertainty threshold (def.: 0.16);
1: $f^{Opt} = $ Clone (parameters $\theta$) of model $f$ {We tune the model anew for every sample / token}
2: Compute logits $f^{Opt}(X) = [f_c^{Opt}(X)]$ and probabilities $p_c(X) = \dfrac{e^{f_c^{Opt}(X)}}{\sum_{k \in C} e^{f_k^{Opt}(X)}}$ for $c \in C$

3: Sort probabilities so that $p_{m(1)}(X) \geq p_{m(2)}(X) \geq \ldots$
4: Compute $\Delta_{1,2} = p_{m(1)}(X) - p_{m(2)}(X)$ {Eq. (1)}
5: **if** $\Delta_{1,2} \geq d_{1,2}$ **then** return **end if**
6: $F := \{m(i) | i \in [1, n_f]\}$
7: **for** $t = 1$ to $T$ **do**    $\theta \leftarrow \theta - \eta \cdot \nabla_\theta L^{Opt}(X)$ {Eq. (2) using focus classes $F$}
8: **return** $\arg\max_{c \in C} f_c^{Opt}(X)$ {Return most likely class}

---

weighting this term cancels for iFo. We discuss conceptual differences to the most common method objective in test time (domain), i.e., minimizing entropy, in the next section.

## 3 THEORETICAL MOTIVATION

We provide intuition by relating our method—optimizing multiple focus classes $F$—to the scenario of optimizing toward a single class, as is common in typical cross-entropy loss computations. Using a simple model and gradient descent equations, we examine which features become more relevant, distinguishing between those shared among classes and those unique to specific classes. We also discuss why a single-step optimization with a larger learning rate can lead to amplification of shared features and be less stable compared to multi-step optimization with a smaller learning rate. Finally, we compare our loss against the prevalent entropy minimization for domain adaptation.

**Model:**   We assume three output classes $C = \{y_0, y_1, y_2\}$ and focus classes $F = \{y_0, y_1\}$. We have four input features $X = \{x_0, x_1, x_2, x_3\}$. The input features $x_j$ can be seen as originating from a prior layer $l$ with parameters $c'$ processing activations $z$, i.e., $x_j = l(z|c')$. Outputs $y_i := f_i(X)$ are computed as:

$$y_0 = c_0 x_0 + c_4 x_3; \quad y_1 = c_1 x_1 + c_5 x_3; \quad y_2 = c_2 x_2 + c_6 x_3 \tag{3}$$

We apply the intuitive notion that a feature value cannot be negative, i.e., $x_i \geq 0$, which happens, for example, after layer activations pass through a non-linearity like $ReLU$. The model implicitly expresses that the presence of input features $\{x_0, x_1, x_2\}$ is only indicative of class $i$, i.e., a change of $x_i$ alters only $y_i$ for $i \in \{0, 1, 2\}$. This implies $c_0 > 0$, $c_1 > 0$, and $c_2 > 0$. The shared feature $x_3$ impacts all outputs $y_i$. It might also be contrastive, i.e., it can be that two parameters from $\{c_4, c_5, c_6\}$ have opposing signs. Depending on the sign of $c_4$, $c_5$ and $c_6$ the presence of the feature ($x_3 > 0$) increases or decreases $y_i$. As we model dependence using shared parameters and activations through $x_3$, we assume that $x_j$ are independent, i.e., rely on different parameters and activations.

**Analysis**   Partial derivatives of the loss with respect to $z_i$:

$$\frac{\partial L^{iFo}}{\partial z_i} = -c_0 \frac{\partial x_0}{\partial z_i} - c_1 \frac{\partial x_1}{\partial z_i} - (c_4 + c_5) \frac{\partial x_3}{\partial z_i}, \quad \frac{\partial L^{doFo}}{\partial z_i} = c_2 \frac{\partial x_2}{\partial z_i} + c_6 \frac{\partial x_3}{\partial z_i} \tag{4}$$

Let us compare the updates for loss $L^{iFo}$ (without weighting) using both focus classes $F = \{y_0, y_1\}$ and using just either $y_0$ or $y_1$, i.e., the loss $L_{y_0,-}$ or $L_{y_1,-}$. Wlog., we use $L_{y_0,-}$.

The changes related to parameters $\{c_0, c_1\}$ tied to a class-specific feature are identical, e.g., $\frac{\partial L^{iFo}}{\partial c_0} = \frac{\partial L_{y_0,-}}{\partial c_0}$. However, the behavior for the shared feature $x_3$ differs between iFo and single class optimization. The feature's impact on outputs can grow or diminish disproportionately: Say both $c_4$ and $c_5$ have the same sign. Assume $c_4 > 0$ and $c_5 > 0$. Then $\frac{\partial L^{iFo}}{\partial z_i} > \frac{\partial L_{y_0,-}}{\partial z_i}$ as $c_4 + c_5 > c_4$. Thus, the change to the shared feature $x_3$ is larger for $L^{iFo}$. In the next update also $c_4$ (and $c_5$) are changed more strongly than for single class optimization as $\frac{\partial L^{iFo}}{\partial c_i} = x_3$ for $i \in \{4, 5\}$. This leads to a "double

growth effect," where the growth of $x_3$ amplifies the growth of parameters $c_4$ and $c_5$ and vice-versa. This could lead to instability in an iterative process, e.g., the shared feature becomes the sole decision criterion with exploding coefficients. This effect is not present if we perform just a single step (with large learning rate). However, for both single and multi-step optimization we observe that the shared feature is altered more. If both $c_4$ and $c_5$ have different signs, i.e., the changes to $x_3$ are less for iFo than for optimizing only $y_0$ and in turn also $c_4$ and $c_5$ change less. Put differently, the relevance of $x_3$ to the decision process diminishes relative to single class optimization. More analysis for other cases is in the appendix.

In summary, ***iFo tends to amplify features that are shared between focus classes*** $F$ (if their presence contributes positively to the likelihood), while doFo tends to hamper features that are shared between focus and out-of-focus classes (as given in the appendix).

**Relation to entropy minimization:** Let us also compare to entropy minimization with loss $L^H = H = -\sum_k p_k \log p_k$. Let us assume that $y_0 = y_1 \geq y_2 \geq 0$ and in turn $p_0 = p_1 \geq p_2$. This is the most important case, as it mimics the situation that the first two classes are very likely but exhibit high uncertainty and the others potentially much less. Thus, changing the outputs comes with smallest possible risk of errors and requires only small updates to the network. Furthermore, if the most likely class is much larger than the second most likely, i.e., in our case $y_0 \gg y_1$, optimization has generally no impact and we do not attempt it.

**Analysis** Partial derivatives with respect to logits $y_i$: $g_k := \frac{\partial H}{\partial y_k} = p_k\big(-H - \log p_k\big)$

With the symmetry $p_0 = p_1$: $g_0 = g_1 =: g_a^{minH} < 0$, $\qquad g_2 =: g_b^{minH} > 0$

Partial derivatives with respect to $c_i$:

$$\frac{\partial L^H}{\partial c_i} = g_a^{minH} x_i \quad i \in \{0,1\}, \frac{\partial L^H}{\partial c_2} = g_b^{minH} x_2, \quad \frac{\partial L^H}{\partial c_i} = g_a^{minH} x_3 \quad i \in \{4,5\}, \frac{\partial L^H}{\partial c_6} = g_b^{minH} x_3$$

The derivatives for entropy minimization $\frac{\partial L^H}{\partial c_i}$ equal those of iFo $\frac{\partial L^{iFo}}{\partial c_i}$ (see Appendix) except for the coefficients $g^{minH}$, which are fixed to 1 for iFo – we introduce the coefficient $\alpha^{iFo} = 1$ for naming purposes. These coefficients determine the strength of updates to parameters and, in turn, the likelihood to change predictions. Figure 3 visualizes the coefficients $\alpha^{iFo}, g_a^{\min H}$ and $g_b^{\min H}$ scaled to 1 for better comparison and as parameter updates and, in turn, coefficients, are multiplied by the learning rate $\eta$, which can be chosen freely. Following our rationale the lowest risk of errors when making changes to the prediction is at 1/3 and 0.5. That is, for $p \approx 0.5$ the uncertainty among the top classes is largest and there are no alternative options. For $p \approx 1/3$ the uncertainty among the top three classes is largest as all have the same probability $p_0 = p_1 = p_2 = 1/3$. In both, choosing any of the most likely classes is reasonable. In between the risk is larger and class changes should be less likely following our rationale. Figure 3 shows that the coefficients emerging from our logits loss follow our rationale better, while entropy is poorly aligned with our rationale. Furthermore, entropy minimization leads to a lower increase of shared features (compared to iFo) (see Appendix). Entropy min. elegantly combines ideas of iFo/doFo (as would SoftMax optimization), i.e., jointly decreases probabilities of unlikely classes (like doFo), while increasing probabilities of likely ones (like iFo).

## 4 EXPERIMENTS

**Setup.** We evaluate our methods on multiple standard vision and language tasks and models with detailed references in the appendix. For our core experiment claiming accuracy improvements of iFo we train on over 70 model-dataset pairs. For other experiments that serve primarily understanding and illustration and doFo (which yielded no clear gains), we relied on fewer pairs. For image classification, we use ImageNet on all pre-trained model types from PyTorch's `torchvision`. If there are multiple model variants differing mostly in size, we aimed for the smallest and the largest model. The full list is shown in Figure 8. The ImageNet dataset is chosen as it is a de-facto standard due to its diversity covering many natural images being evaluated on a rich set of diverse pre-trained architectures. For language modeling, we use from HuggingFace: GPT-2 (124MB version), Llama 3.2 1B, QWEN 2.5 1.5B, Fox-1 1.6B, StableLM 1.6B, and Gemma-3 1B. We evaluate on OpenWebText (an open-source replication of the WebText dataset from OpenAI), SimpleWiki (Wikipedia articles written in simple

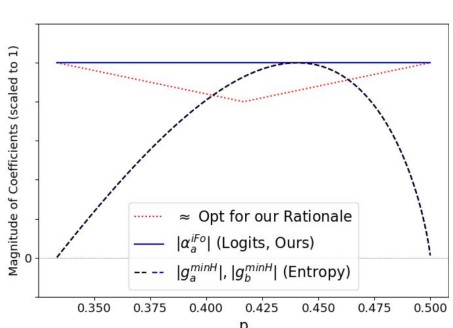

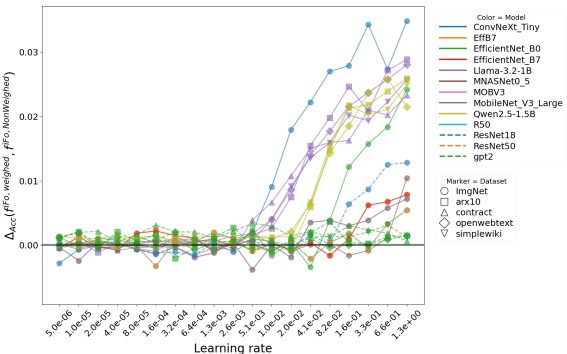

Figure 3: Coefficients for entropy-based minimization are poorly aligned with our rationale.

Figure 4: Differences in accuracies for $f^{iFo}$ using probability-weighted ($p_c$) or non-weighted likely class terms in the loss function (Eq. 2).

English), legal contracts, Arxiv abstracts, bookcorpus and CNN Dailymail (news articles) taking chunks of 128 tokens and predicting the last token given all prior ones. Our datasets cover output classes ranging from about 1000 to more than 100000 and different text domains and styles to ensure generalizability. For hyperparameters, if not otherwise stated, we use defaults specified in Algorithm 1. Regarding the optimizer, we chose SGD without weight decay as discussed in the Appendix.

**Measures and Notation.** We compare the model $f^{Opt}$ resulting from our optimization (Algorithm 1) against the original, unmodified model $f$ serving as baseline. We consider various configurations $O = O_{Img} \cup O_{Text} = \{f, D\}$, meaning pairs of models and datasets $(f, D)$ for images $O_{Img}$ and text $O_{Text}$. We report measures for individual configurations $(f, D) \in O$ and a test dataset $D_{te} = \{X, Y\} \subset D$ and aggregated measures on configurations $O$. The subset $D_{te,1,2} \subset D_{te}$ is the set of uncertain samples, where optimization takes place, i.e., $D_{te,1,2} = |\{(x, y) \in D_{te} | \Delta_{1,2} < d_{1,2}\}|$. Thus, the fraction of samples for which optimization took place is $\frac{|D_{te,1,2}|}{|D|}$. For each $O_{Text}$ we identified uncertain samples ($\Delta_{1,2} < d_{1,2}$) until we had $|D_{te,1,2}| = 20000$ yielding a variable dataset size $|D|$ per configuration. For each $O_{Img}$ we filtered the full validation dataset $|D| = 50000$ yielding variable sized $|D_{te,1,2}|$ per configuration as obtaining $|D_{te,1,2}| = 20000$ was not possible for small thresholds $d_{1,2}$. The accuracy of model $f'$ is

$$acc(f') = \frac{|\{x | (\arg\max_c f'_c(x)) = y, (x, y) \in D_{te,1,2}\}|}{|D_{te,1,2}|}$$

That is, we only evaluate models on the uncertain samples $D_{te,1,2}$, as for all other samples $D \setminus D_{te,1,2}$ we do not employ our method. $\Delta_{Acc}(f'', f') := acc(f'') - acc(f')$ refers to the gain, i.e., the difference in accuracy of $f''$ and $f'$, where $f''$ is mostly the optimized model using iFo, i.e., $f'' = f^{iFo}$ and $f'$ the original model, i.e., $f' = f$. We use the overline to indicate averages, i.e., $\overline{\Delta_{Acc}(f'', f')}$ is the average accuracy $\Delta_{Acc}$ across a set of configurations. We also compute the number of configurations from $O'$, where model $f''$ yielded accuracy gains compared to $f'$:

$$\#\Delta_{Acc>0} := |\{(f, D) \in O' | \Delta_{Acc}(f'', f') > 0\}|$$

**Uncertainty Assessment.** *Experiment 1 - Top-k Accuracy vs. Threshold $d_{1,2}$:* First, we assess implicit assumptions that make our method more or less likely to be beneficial. We investigate the impact of our uncertainty assessment on the top-k accuracy, i.e., a network's output is correct as long as the correct class is among the $k$ most likely classes. We want that even for high uncertainty (i.e., when keeping only samples with small differences $d_{1,2}$) the top-k accuracy rapidly increases for $k$. The top-2 accuracy should be significantly higher than the top-1 accuracy and the top-1 accuracy should be low. If the top-1 accuracy is 0 and top-2 accuracy is 1 then our method can only improve predictions with two focus classes $n_f = |F| = 2$. The ideal sample has only two likely candidate classes, both with probability 0.5 so that (for well-calibrated models) on average accuracy would show a step from 0.5 to 1 for k=1 to k=2 and remain at 1. We compute the top-k accuracy on datasets $D_{te,1,2}$ using multiple uncertainty levels, i.e., very high ($d_{1,2} = 0.04$), high ($d_{1,2} = 0.16$), and very low ($d_{1,2} = 0.84$).

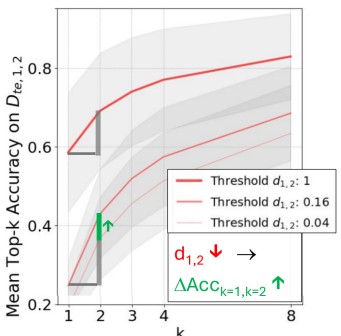

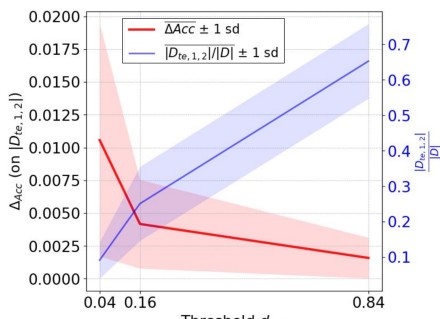

Figure 5: Difference in Mean Top-k Accuracy $\Delta\text{Acc}_{k=1,k=2}$ for non-tuned models $f$ across all configurations $O$ grows with lower threshold $d_{1,2}$.

Figure 6: Larger threshold $d_{1,2}$ implies more samples are uncertain and optimized($\frac{|D_{te,1,2}|}{|D|}$ grows). But the relative gain in terms of accuracy $\Delta_{Acc}(f^{iFo}, f)$ on uncertain samples $D_{te,1,2}$ declines.

*Result:* In Figure 5 we observe that for samples with lower prediction uncertainty (lower $d_{1,2}$) the top-k likelihood also gets smaller, i.e., the correct class is less likely among the top-k correct classes. This is expected as uncertain samples appear more difficult to classify. Furthermore, we observe that top-k accuracy considerably grows with $k$ for all $d_{1,2}$. But its growth is strongest for uncertain samples (small $d_{1,2}$). The plot hints that applying our method only on high uncertainty samples (small $d_{1,2}$) is favorable in multiple ways: First, the chances that changes to models lead to an incorrect prediction are lower, as the odds of the most likely class being incorrect before applying our method are already high. Second, the top-1 accuracy is lower and the change from top-1 to top-2 is larger than for low uncertainty. That is, the slope from $k = 1$ to $k = 2$ tends to increase with smaller $d_{1,2}$. Third, we do not need to change probabilities a lot. This seems an easier, more local optimization task.

Curves for individual models vary significantly, as indicated by the standard deviation (grey area in Figure 5). For language models (individual models shown in Appendix in Figure 20), the curves for lower uncertainty levels ($d_{1,2} \in \{0.04, 0.16\}$) are flatter than for image models. Flatter curves indicate that fixing a prediction is harder as there are more candidate classes with a comparable likelihood. However, the top-1 accuracy for language models is also lower, meaning that changing the prediction is likely not harmful (as the most likely class is more likely wrong). Thus, overall, at the outset it is not clear, for which model and dataset combination our method should perform best. All configurations seem reasonable candidates to test our method, as they fulfill the basic criteria that top-2 accuracy is significantly higher than top-1, while top-1 accuracy is not very high for low certainty.

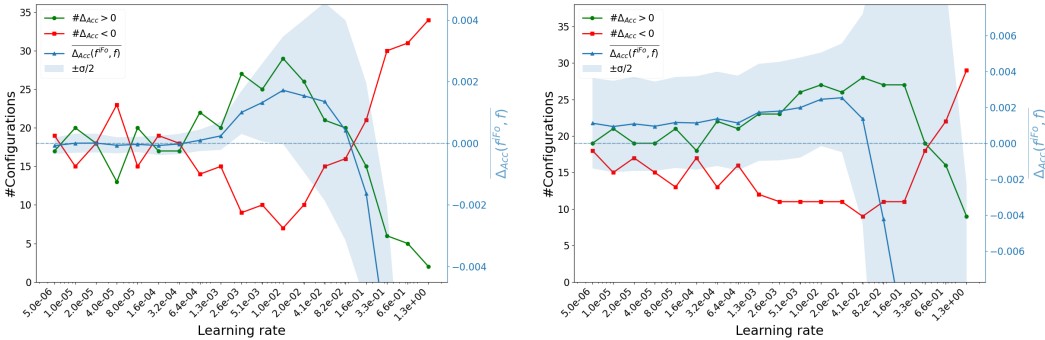

Figure 7: Average $\Delta_{Acc}$ and $\#\Delta_{Acc} > 0$ for different learning rates for all configurations $O_{Text}$

Figure 8: Average $\Delta_{Acc}$ and $\#\Delta_{Acc} > 0$ for different learning rates for all configurations $O_{Img}$

**Hyperparameters and Performance.** *Experiment 2 - Learning rate $\eta$ and performance:* Figures 7 and 8 show that for small learning rates neither significant gains nor losses are observed on average and the number of classifiers that benefit from using iFo is about equal to those that do not, while for very large learning rates iFo hurts performance. In between we find that both the average benefit ($\overline{\Delta_{Acc}(f^{iFo}, f)}$) as well as the number of configurations with $\Delta_{Acc} > 0$ increase with the learning

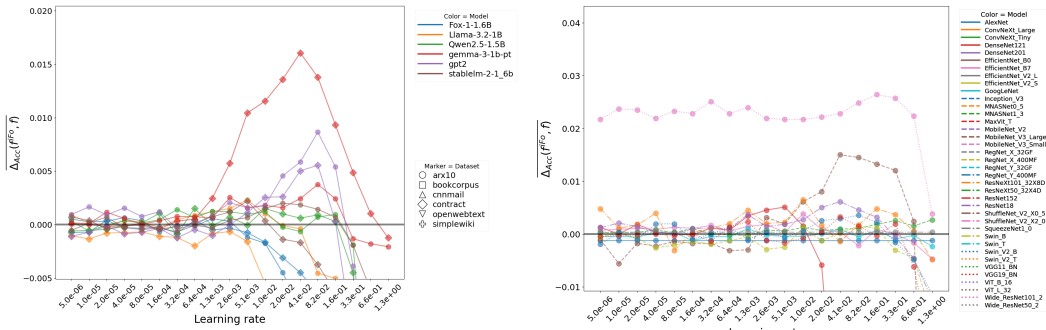

Figure 9: $\Delta_{Acc}$ across learning rates for 1/3 of $O_{Text}$ configurations (all in Appendix in Fig. 12)

Figure 10: $\Delta_{Acc}$ across learning rates for 1/3 of $O_{Img}$ configurations (all in Appendix in Fig. 11)

rate up to some point. To show significant of these gains, let us first investigate text configurations (Figure 7). We see that for a number of sequential learning rates, i.e. 4, we have 24 or more successes ($\Delta_{Acc} > 0$) and 10 or less losses ($\Delta_{Acc} < 0$). A one-sided binomial-test reveals using the lower bound on successes (25) and upper bound on losses (10) a p-value of 0.0059 that there are more successes than losses. For images (Figure 8), we have four consecutive learning rates a lower bound of 26 successes and an upper bound of 11 losses, given a p-value of 0.0066. Thus, there is strong evidence for both text and image models that iFo is beneficial for the majority of models in both modalities. Aside from testing whether the majority of models have positive gains, we also tested if overall the mean gain across all configurations is $\overline{\Delta_{Acc}} = 0.28$ with standard deviation 0.55 across all models is larger than 0 using Table 4. This was confirmed with p-value 0.004 and confirms that there are no extreme outliers with low performance. Furthermore, when testing on individual decisions, i.e.,

To further illustrate, we also plotted a subset of all individual text and image configurations in Figures 9 and 10. While most text and image models follow the aggregate pattern there are a number of positive and negative exceptions, e.g., for gpt2 on bookcorpus almost no gains exist, while for gemma-3 (on most datasets) gains are large. There are also three configurations that appear erroneous, i.e., WideResNet (which is shifted up - see pink line in Figure 10) and AlexNet, which gives the exact same (negative) gain. However, removing them does not change the significance of our results, but it would considerably lower the mean and standard deviation on the $\Delta_{Acc}$ (Figure 10), so that they appear more similar to text configurations (Figure 9). We shall discuss exceptions more in the appendix.

*Experiment 3 - Uncertainty threshold $d_{1,2}$:* We assess our method by varying the uncertainty threshold $d_{1,2}$ for a fixed learning rate. We used one learning rate across all more than 30 image models and per model learning rates for LLMs as given in the Appendix in Table 4. To this end, we show the difference in terms of accuracy between the unmodified baseline model $f$ and the tuned model $f^{iFo}$ (on the uncertain samples $D_{te,1,2}$). We also show the fraction of uncertain samples $\frac{|D_{te,1,2}|}{|D|}$ depending on the uncertainty threshold $d_{1,2}$.

*Results* in Figure 6 show that with larger uncertainty threshold $d_{1,2}$ the fraction of uncertain samples also increases $\frac{|D_{te,1,2}|}{|D|}$, while the difference in accuracy $\Delta_{Acc}$ decreases. As computation grows with the number of uncertain samples $|D_{te,1,2}|$, the declining $\Delta_{Acc}$ (on $|D_{te,1,2}|$) shows that the gain per computation, e.g., FLOP, declines with larger $d_{1,2}$. This might even hold although for larger $d_{1,2}$ overall more samples might be corrected, but the growth in correct samples is less than that of $|D_{te,1,2}|$. Figure 6 also shows that the standard deviation for $\Delta_{Acc}$ decreases with bigger $d_{1,2}$, which is expected due to the law of large numbers as also $|D_{te,1,2}|$ grows and, thus, we evaluate on a larger dataset. (For individual dataset-model pairs see Figure 21 in Appendix.)

**Ablation.** *Experiment 4 - Impact of weighting:* Figure 4 shows the difference in accuracy when weighting losses with softmax probabilities $p_c$ versus non-weighting (see Equation 2). We can see that weighting has a somewhat positive, but modest impact in the noisy area where the learning rate is low and generally few predictions are changed due to focusing on the likely classes. The impact increases for larger learning rates where more changes occur. For large learning rates, gains can be substantial. However, weighting does not yield gains in all cases.

**Comparison to related techniques.** *Experiment 5 - iFo vs. domain adaptation techniques* Techniques addressing the same problem are Tent, PASLE, and SAR (see Table 1). We tuned the learning rate and reset parameters after each sample according to our problem definition, conducted just one optimization step and evaluated on the same set of samples (e.g., those deemed uncertain $d_{1,2} = 0.16$) for Tent and SAR. For PASLE, we used its own mechanism, ensuring that the number of uncertain samples roughly corresponds to those of our method. We ran on regular ImageNet consisting of in-domain data using configurations $O_{Img}$. More details including a discussion of wall-clock time are in the Appendix A.3.2.

*Results* in Table 2 show that Tent and PASLE both lead to statistically significant gains like iFo, while SAR does not. For SAR it seems best to use a learning rate of 0, where no predictions are changed. SAR minimizes entropy with a few extras, which seem to hurt performance (also relative to Tent which does better). However, compared to our method $iFo$, all methods perform significantly worse. Tent minimizes entropy, which might lead to gains according to our theoretical motivation, but is conjectured to be lower. Tent optimizes only affine parameters of (normalization) layers, which might also limit (or enhance) its gains. Our ablation replacing our optimization objective with entropy suggests that both reasons seem relevant. PASLE also minimizes entropy but only on the set of likely classes, which is determined dynamically and on average there are three or more (depending on the threshold $\tau(r)$), while we use two. As explained in Section 2 this leads to the situation that (i) if all classes are equally uncertain, no optimization takes place and (ii) if a feature is shared among more classes (e.g., all instead of just the top 2) optimization might lead to a lower increase. Both of which are against our rationale.

Table 2: Comparison to other methods on ImageNet, i.e., $O_{Img}$. IFo performs best.

| Method | lr | $\overline{\Delta_{Acc}}$ | Std $\Delta_{Acc}$ | pval $\overline{\Delta_{Acc}^{iFo}} > \overline{\Delta_{Acc}}$ | pval $\overline{\Delta_{Acc}} > 0$ |
|---|---|---|---|---|---|
| SAR | 3.1e-06 | -0.0011 | 0.0109 | 0.0019 | 0.73 |
| Tent | 1.0e-04 | 0.02 | 0.04 | 0.0019 | 0.01 |
| PASLE | 5.0e-05 | 0.13 | 0.77 | 0.0020 | 0.16 |
| **Ours, iFo** | 2.05e-02 | 0.28 | 0.55 | - | 0.004 |

*Further experiments and analysis* can be found in the Appendix.

Table 3: Methodological comparison with adaptation methods.

| Aspect | PASLE | Tent | ReCAP | SAR | **Ours** |
|---|---|---|---|---|---|
| Loss Function | Custom | Entropy | (localized) Entropy | (reliable,flat) Entropy | Logits/Cross-Ent. |
| Parameters optimized | All | Norm. Layers | Norm. Layers | Norm. Layers | All |
| Pseudo-labels | Hard, custom loss | - | - | - | Hard, soft weighting |
| Sample splitting | ✓ | ✗ | ✗ | ✓ | ✓ |
| Optimize only subset | ✗ | ✗ | ✗ | ✓ (certain) | ✓ (uncertain) |
| No extra needs | ✓ | ✗ (augment) | ✗ (in-domain data) | ✓ | ✓ |

# 5 RELATED WORK

Studies on domain adaptation (see Table 1 and others (Kim et al., 2024; Osowiechi et al., 2024; Karmanov et al., 2024)) assume that test-time data originate from a domain different from that of the training data, whereas we focus on in-domain samples. Closest to our work in terms of problem setup is MEMO. Its main focus is domain adaptation, but also stresses robustness more broadly and it also evaluates on in-domain samples. However, its gains for in-domain samples rely only on augmentations rather than on the proposed entropy minimization. Furthermore, methods that do evaluate on in-domain samples such as MEMO present only few dataset/model configurations without showing gains. A key conceptual differentiation is the optimization. Our method uses logits, whereas most others use some form of entropy minimization.

Methodologically, PASLE and SAR (Hu et al., 2025a; Niu et al., 2023) share most ideas with our method (see Table 3). Some overlaps such as sample splitting appear coincidental as they serve different purposes or lead to opposing decisions. SAR optimizes only certain samples (measured by entropy), whereas we only optimize on uncertain samples (measured by differences in softmax probs). PASLE conceptually also splits samples into uncertain and certain but optimizes on all. Our motivation for splitting is to save on unnecessary computation, while PASLE's motivation is higher accuracy. PASLE, among other works in TTA, investigated focusing on more than one target class, e.g., hard labels for likely classes (Hu et al., 2025a) and entropy minimization (Wang et al., 2020; Zhang et al., 2022; Hu et al., 2025b). PASLE's optimization of uncertain samples can be seen as

saying: The model needs no change if the probability mass is concentrated among all likely classes. For example, if two classes $a, b$ for a sample both have probability $p_a = p_b = 0.5$, PASLE's loss is $\approx \log(p_a + p_b) = \log(1) = 0$, while our loss in this case is far from 0. That is, we take the opposite stance of PASLE as we deem our method to be most promising in this situation. We provide details in the Appendix. Instead of framing our approach in terms of distributions, we assume that an individual sample might not be classified perfectlyand do not aim at continual learning. In turn, while works in domain-adaptation often only optimize scaling and shifting of normalization layers, we optimize all model parameters. Numerous other studies have explored test-time adaptation (surveyed in Liang et al. (2024)). To our knowledge, almost all prior works (see Table 1) have extra needs beyond the test sample. They require additional tasks (Liu et al., 2021; Sun et al., 2020) or data (potentially, self-generated using augmentations (Cotta, Memo, DeYO (Wang et al., 2022; Zhang et al., 2022; Lee et al., 2024)) or in-domain data (Liu et al., 2021) or in-domain samples(Hu et al., 2025b)). We intentionally focus on just a single test sample and refrain from such extra needs as they limit practical applicability, require effort, and often pose risks. Common augmentations such as flipping and rotation can be fatal on trivial datasets such as MNIST, e.g., label confusions due to 180 degree rotations (6 becomes 9). Many techniques focusing on domain-adaptation are complementary to our approach and could potentially be integrated to enhance results. Commonly, test-time tuning relies on nearest neighbors and conducts a form of local learning by adjusting the model to the chosen data such as TSD+MLSC (Wang et al., 2023) and others (Sun et al., 2024; Bottou & Vapnik, 1992; Hardt & Sun, 2023; Hübotter et al., 2024). (Mummadi et al., 2021) aims at accounting for distribution shifts using the soft likelihood ratio (SLR) loss. Conceptually, $iFo$ and $doFo$ are multi-class optimizations, where we consider all focus (and out-of-focus) classes as target classes. Our technique is reflective as it often does not yield a decision based on a single forward pass (Zhong et al., 2024; Schneider, 2025; Madaan et al., 2024; Schneider & Vlachos, 2024; Selvaraju et al., 2017). When tuning embedding vectors (rather than the entire network), our technique can also be seen through the lens of moving the embedding vectors in a direction (i.e., that of shared representations (for iFo)). This interpretation is well-known for word vectors with some evidence (Mikolov et al., 2013) for more abstract function vectors in LLMs (Todd et al., 2024).

## 6 LIMITATIONS, DISCUSSION AND CONCLUSIONS

This work set out to answer the research question *Does focusing on likely classes of a single, in-domain sample improve model predictions?* by aiming for a positive confirmation proposing two optimization methods. Our theoretical model enables concise statements and offers deeper insights into the proposed methods; however, it lacks general theorems necessary to conclusively answer the research question. Our empirical findings suggest that the assumption that reducing features of unlikely classes is beneficial does not hold consistently, as indicated by the outcomes for doFo. This may be due to our hyperparameter choices—specifically, aggressively reducing shared features across all but the two most likely classes. Furthermore, doFo unlearns associations, which is tricky, e.g., literature on unlearning indicates that unlearning facts can also inhibit general laws. However, our assumption that features belonging to classes with low probabilities are less useful might also be incorrect. It seems more to be the case that non-shared features among (non-likely) classes should be reduced as they might be less robust as the fact that a feature is "shared" provides support for its reliability. In contrast, enhancing shared features of likely classes (iFo) proves to be beneficial. This yields the conjecture that *reducing reliance on a few strongly activated features is beneficial during regular offline training (as witnessed by techniques such as dropout and weight decay), whereas enhancing such features can be advantageous at test time.*Our feature-level perspective aligns with the contrasting objectives of training phases: broad generalization during offline training and targeted local adaptation during online, test-time training. We empirically demonstrated, supported by theoretical reasoning, that performing a single gradient step (with high learning rates) can effectively replace multiple gradient computations in our setup. This makes our method more practical, though the computational overhead is still considerable.Additionally, our method uses uncertainty estimates. We leveraged the standard notion of softmax probabilities of neural networks, which are often poorly calibrated (Guo et al., 2017) though not so much for pre-trained LLMs (Xie et al., 2024). Our method inherently depends on differences in network outputs as changes are more common if the differences between softmax probabilities are small. In such a case, the network does not need to be changed much to yield different outcomes. Applying calibration techniques might improve results.

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

# A APPENDIX

## A.1 OPTIMIZING SOFTMAX OUTPUTS INSTEAD OF LOGITS

Using logs of softmax outputs:

$$\textbf{Given:} \quad \mathbf{z} = (z_1, z_2, \ldots, z_K), \quad p_i = \frac{e^{z_i}}{\sum_{j=1}^{K} e^{z_j}},$$

$$\mathbf{y} = (y_1, y_2, \ldots, y_K) \text{ with } y_c = 1 \text{ and } y_i = 0 \text{ for } i \neq c.$$

**Cross-entropy loss:**

$$L = -\sum_{i=1}^{K} y_i \log p_i = -\log p_c = -\log\Big(\frac{e^{z_c}}{\sum_{j=1}^{K} e^{z_j}}\Big).$$

Simplifying:

$$L = -\log\Big(\frac{e^{z_c}}{\sum_{j=1}^{K} e^{z_j}}\Big) = -\big[\log(e^{z_c}) - \log(\sum_{j=1}^{K} e^{z_j})\big] = \log(\sum_{j=1}^{K} e^{z_j}) - z_c.$$

Neglect any weighting and assume we have focus classes $F$ to optimize towards 1; this yields (as done for iFo):

$$L = 1/|F| \sum_{c \in F} (\log\Big(\sum_{j=1}^{K} e^{z_j}\Big) - z_c) = \log\Big(\sum_{j=1}^{K} e^{z_j}\Big) - 1/|F| \sum_{c \in C} z_c.$$

We push each $z_j \notin C$ towards minus $\infty$, while pushing the others towards positive $\infty$.

Assuming we have classes $F \setminus C$ to optimize towards 0 this yields:

$$L = 1/|F \setminus C| \sum_{c \in F \setminus C} z_c - 1/|F \setminus C| \sum_{c \in F \setminus C} (\log\Big(\sum_{j=1}^{K} e^{z_j}\Big) - z_c) = 1/|F \setminus C| \sum_{c \in F \setminus C} \log\Big(\sum_{j=1}^{K} e^{z_j}\Big).$$

We push each $z_j$ towards minus $\infty$, while pushing the others towards positive $\infty$.

Thus, comparing optimizing raw logits vs softmax outputs, we see that raw logits allow us to focus more directly only on the intended classes by altering their outputs, while softmax impacts directly all $K$ classes as seen by the summations. For our doFo approach (with softmax outputs) the normalization term increases classes, while for iFo (with Softmax) they are we decreased. In that sense, softmax yields a combined approach.

## A.2 MORE DETAILS ON EXPERIMENTAL SETUP

**Motivation for SGD rather than Adam and no weight decay.** We use SGD and not Adam, as Adam relies on moments that anticipate future or non-local changes—effectively hinting at how parameter updates might evolve when moving in a certain direction across iterations. Two properties of our method make this less relevant. First, we chose to use fixed focus classes for a single sample throughout the optimization. Second, we use mostly just a single step for stability and computational reasons. Thus, we expect limited variation in the optimization direction across iterations and anticipate that upcoming iterations provide little benefit. Following similar reasoning, we set momentum to 0, since momentum primarily prevents zig-zagging and escaping local minima by averaging gradients across multiple batches and iterations. We also set weight decay to 0, as weight decay penalizes large weights, causing the network to rely on many features. However, our philosophy is the opposite: We aim to focus on fewer relevant classes and features. Therefore, weight decay might counteract our method.

**Hardware and Software.** All experiments were conducted on an Ubuntu 22.04 system equipped with Python 3.12, PyTorch 2.5, CUDA 12.8, and multiple NVIDIA H100s and an NVIDIA RTX 4090 GPU.

**Datasets and Models.** References for datasets and models: For image classification, we used Pytorch's torchvision models v0.22 (`https://docs.pytorch.org/vision/stable/models`) trained on ImageNet1K_V1 (Deng et al., 2009) (if available). We used IMAGENET1KV1 versions as they are most widely supported among models.

For language modeling, we use (from HuggingFace `https://huggingface.co/`, accessed May 2025): GPT-2 (Radford et al., 2019) 124M (version), Llama 3.2 1B (meta-llama/Llama-3.2-1B) (Touvron et al., 2024), QWEN 2.5 1.5B (Qwen/Qwen2.5-1.5B) (Chen et al., 2024), Gemma-3 1B(google/gemma-3-1b-pt) (Team, 2025), stable LM (stabilityai/stablelm-2-1_6b)(Bellagente et al., 2024), and Fox-1 1.6B (tensoropera/Fox-1-1.6B)(Hu et al., 2024). We evaluate on datasets mostly from huggingface namely OpenWebText and open-source replication of the WebText dataset from OpenAI (Gokaslan & Cohen, 2024) (Skylion007/openwebtext), SimpleWiki, a collection Wikipedia articles written in simple English (Wikipedia, 2024) (from `https://dumps.wikimedia.org/simplewiki/20250501/`), legal contracts (del Moral, 2024) (albertvillanova/legal_contracts), Arxiv-10 (effectiveML/ArXiv-10) (Farhangi et al., 2022), bookcorpus (Zhu et al., 2015) (rojagtap/bookcorpus), and CNN DailyMail (See et al., 2017) (abisee/cnn_dailymail Config 3.0.0).

For pretrained ImageNet networks, we relied on the standard validation data of 50000 samples. For text-data we used only pre-trained models and, thus, we did not perform any train/test split. Like for any public dataset, we cannot exclude that the models were already trained on the data. However, one reason for using up to 2B models was that they were far from perfect on this data, which is essential for our work. Therefore, the fact that the data might have been used for training is not a big concern.

## A.3 ADDITIONAL EXPERIMENTS AND FURTHER COMMENTS ON EXPERIMENTS

### A.3.1 AD EXPERIMENT 2 - PERFORMANCE OF INDIVIDUAL MODELS (FOR FIXED HYPERPARAMETERS)

Here we show accuracy differences of our optimized models $f^{iFo}$ compared to the non-optimized models $f$ for fixed hyperparameters, in particular the learning rate. This is in addition to figures showing data for varying learning rates, i.e., Figures 8 and 7 that show aggregate data, and Figures 12 and 11 that show all individual configurations. To fix the learning rate, we opted for a simple approach that aims at finding just one learning rate for all configurations $O_{Img}$, while we chose network dependent learning rates for $O_{Text}$ (but the same rate for all datasets). Clearly, gains would be larger if we chose learning rates per model-dataset pair. However, as we chose learning rates based on the test data, we want to minimize any information leakage. For text configurations, we picked the learning rate that overall gave the largest average gain, i.e., peaks for $\Delta_{Acc}$ in Figures 8 and 7. This is not necessarily the learning rate that is beneficial across the largest set of models but it tends to be close to it as shown in the figures. Table 4 shows key metrics for $O_{Img}$ and $O_{Text}$. For once, we see that the learning rate giving best performance for text models varies between roughly 1e-2 and 5e-3, which is a considerable gap. For our fixed uncertainty threshold $d_{1,2} = 0.16$ and LLMs we note that the fraction of uncertain samples varies considerably across datasets varies by more than 30% (ranging from roughly 25% for contracts to 60% for bookcorpus), while the variance for a fixed dataset across model is generally mostly within a 10% difference (except for GPT-2 which is still within 15%). The same cannot be said for image classifiers, where for the same dataset the fraction varies also up to 30%. This is not unexpected as LLMs all have similar architectures (all being transformer variants), while image classifier have much more diverse architecture including transformers as well as convolutional neural networks. Model size seems not to be a key factor for whether our method is beneficial as can be best seen by focusing on image models, e.g., for SWIN base models lose, while tiny models gains and for ShuffleNet it is opposite. When it comes to architectural elements we could not identify strong indicators, e.g., width or the presence of residual networks appear to be highly beneficial in some cases (like WideResNet) but not always. However, for WideResNet we also observed strange behavior, e.g., there seems to be an offset as seen in Figure 10, i.e. the curve shows characteristics of other classifiers (e.g., with a steep drop in the end and

| Model | Dataset | $\frac{|D_{te,1,2}|}{|D|}$ % Uncertain Sa. | $|D_{te,1,2}|$ #Uncertain Sa. | lear. rate | **Acc.**$f$ Baseline | **Acc.**$f^{Opt}$ Ours, iFo | $\Delta_{Acc}$ |
|---|---|---|---|---|---|---|---|
| Fox-1-1.6B | arx10 | 39.7 | 20000 | 2.56e-03 | 19.62 | 19.65 | 0.03 |
| Fox-1-1.6B | bookc | 64.6 | 20000 | 2.56e-03 | 12.36 | 12.57 | 0.21 |
| Fox-1-1.6B | cnnma | 41.9 | 20000 | 2.56e-03 | 20.82 | 21.04 | 0.22 |
| Fox-1-1.6B | contr | 29.0 | 20000 | 2.56e-03 | 23.42 | 23.38 | -0.03 |
| Fox-1-1.6B | openw | 41.2 | 20000 | 2.56e-03 | 20.26 | 20.4 | 0.15 |
| Fox-1-1.6B | simpl | 36.3 | 20000 | 2.56e-03 | 20.24 | 20.29 | 0.05 |
| Llama-3.2-1B | arx10 | 37.4 | 20000 | 2.56e-03 | 21.22 | 21.37 | 0.15 |
| Llama-3.2-1B | bookc | 61.7 | 20000 | 2.56e-03 | 13.82 | 14.01 | 0.18 |
| Llama-3.2-1B | cnnma | 40.6 | 20000 | 2.56e-03 | 21.58 | 21.58 | -0.0 |
| Llama-3.2-1B | contr | 25.6 | 20000 | 2.56e-03 | 24.48 | 24.41 | -0.06 |
| Llama-3.2-1B | openw | 40.1 | 20000 | 2.56e-03 | 20.04 | 20.08 | 0.05 |
| Llama-3.2-1B | simpl | 33.7 | 20000 | 2.56e-03 | 20.42 | 20.57 | 0.15 |
| Qwen2.5-1.5B | arx10 | 35.0 | 20000 | 1.02e-02 | 20.4 | 20.52 | 0.12 |
| Qwen2.5-1.5B | bookc | 50.9 | 20000 | 1.02e-02 | 15.06 | 15.2 | 0.15 |
| Qwen2.5-1.5B | cnnma | 39.0 | 20000 | 1.02e-02 | 20.74 | 20.82 | 0.07 |
| Qwen2.5-1.5B | contr | 21.3 | 20000 | 1.02e-02 | 25.95 | 26.1 | 0.15 |
| Qwen2.5-1.5B | openw | 39.9 | 20000 | 1.02e-02 | 20.3 | 20.37 | 0.07 |
| Qwen2.5-1.5B | simpl | 32.9 | 20000 | 1.02e-02 | 20.76 | 20.88 | 0.12 |
| gemma-3-1b-pt | arx10 | 33.0 | 20000 | 4.10e-02 | 17.25 | 17.46 | 0.21 |
| gemma-3-1b-pt | bookc | 61.3 | 20000 | 4.10e-02 | 13.44 | 13.42 | -0.03 |
| gemma-3-1b-pt | cnnma | 33.1 | 20000 | 4.10e-02 | 17.08 | 18.76 | 1.68 |
| gemma-3-1b-pt | contr | 21.1 | 20000 | 4.10e-02 | 19.71 | 21.09 | 1.39 |
| gemma-3-1b-pt | openw | 31.5 | 20000 | 4.10e-02 | 16.23 | 17.46 | 1.23 |
| gemma-3-1b-pt | simpl | 27.6 | 20000 | 4.10e-02 | 17.64 | 19.45 | 1.82 |
| gpt2 | arx10 | 37.9 | 20000 | 4.10e-02 | 18.88 | 19.47 | 0.59 |
| gpt2 | bookc | 52.6 | 20000 | 4.10e-02 | 10.51 | 10.46 | -0.04 |
| gpt2 | cnnma | 33.0 | 20000 | 4.10e-02 | 19.63 | 19.78 | 0.15 |
| gpt2 | contr | 26.8 | 20000 | 4.10e-02 | 21.4 | 21.92 | 0.52 |
| gpt2 | openw | 31.7 | 20000 | 4.10e-02 | 19.28 | 19.36 | 0.08 |
| gpt2 | simpl | 33.9 | 20000 | 4.10e-02 | 19.72 | 19.97 | 0.25 |
| stablelm-2-1_6b | arx10 | 37.9 | 20000 | 1.02e-02 | 20.04 | 20.22 | 0.18 |
| stablelm-2-1_6b | bookc | 58.2 | 20000 | 1.02e-02 | 16.77 | 17.09 | 0.32 |
| stablelm-2-1_6b | cnnma | 38.4 | 20000 | 1.02e-02 | 21.49 | 21.58 | 0.09 |
| stablelm-2-1_6b | contr | 19.3 | 20000 | 1.02e-02 | 27.0 | 27.07 | 0.07 |
| stablelm-2-1_6b | openw | 39.4 | 20000 | 1.02e-02 | 21.22 | 21.28 | 0.07 |
| stablelm-2-1_6b | simpl | 31.6 | 20000 | 1.02e-02 | 22.2 | 22.28 | 0.07 |
| AlexNet | Image | 31.3 | 15630 | 2.05e-02 | 23.25 | 23.13 | -0.12 |
| ConvNeXt_Large | Image | 6.9 | 3449 | 2.05e-02 | 37.75 | 37.46 | -0.29 |
| ConvNeXt_Tiny | Image | 11.3 | 5629 | 2.05e-02 | 37.54 | 36.99 | -0.55 |
| DenseNet121 | Image | 14.1 | 7029 | 2.05e-02 | 31.11 | 30.53 | -0.58 |
| DenseNet201 | Image | 11.7 | 5867 | 2.05e-02 | 31.52 | 31.38 | -0.14 |
| EfficientNet_B0 | Image | 14.5 | 7241 | 2.05e-02 | 33.75 | 34.18 | 0.43 |
| EfficientNet_B7 | Image | 11.0 | 5502 | 2.05e-02 | 28.44 | 28.54 | 0.09 |
| EfficientNet_V2_S | Image | 8.3 | 4148 | 2.05e-02 | 34.47 | 34.47 | 0.0 |
| GoogLeNet | Image | 23.9 | 11952 | 2.05e-02 | 31.43 | 31.44 | 0.01 |
| Inception_V3 | Image | 6.6 | 3308 | 2.05e-02 | 22.4 | 22.61 | 0.21 |
| MNASNet0_5 | Image | 28.0 | 13992 | 2.05e-02 | 30.72 | 30.88 | 0.16 |
| MNASNet1_3 | Image | 36.2 | 18122 | 2.05e-02 | 49.54 | 49.6 | 0.07 |
| MaxVit_T | Image | 6.8 | 3424 | 2.05e-02 | 37.41 | 36.95 | -0.47 |
| MobileNet_V2 | Image | 16.2 | 8082 | 2.05e-02 | 29.3 | 29.44 | 0.14 |
| MobileNet_V3_Large | Image | 11.4 | 5718 | 2.05e-02 | 28.05 | 28.86 | 0.8 |
| MobileNet_V3_Small | Image | 20.0 | 9989 | 2.05e-02 | 25.71 | 26.23 | 0.52 |
| RegNet_X_32GF | Image | 6.7 | 3343 | 2.05e-02 | 32.25 | 32.87 | 0.63 |
| RegNet_X_400MF | Image | 14.5 | 7245 | 2.05e-02 | 30.46 | 30.34 | -0.12 |
| RegNet_Y_32GF | Image | 6.1 | 3062 | 2.05e-02 | 32.14 | 32.2 | 0.07 |
| RegNet_Y_400MF | Image | 13.5 | 6754 | 2.05e-02 | 30.77 | 31.12 | 0.36 |
| ResNeXt101_32X8D | Image | 7.1 | 3530 | 2.05e-02 | 32.44 | 32.61 | 0.17 |
| ResNeXt50_32X4D | Image | 9.2 | 4601 | 2.05e-02 | 32.04 | 33.08 | 1.04 |
| ResNet152 | Image | 9.6 | 4795 | 2.05e-02 | 31.91 | 32.7 | 0.79 |
| ResNet50 | Image | 12.2 | 6087 | 2.05e-02 | 31.25 | 32.18 | 0.94 |
| ShuffleNet_V2_X0_5 | Image | 24.0 | 12013 | 2.05e-02 | 22.33 | 22.94 | 0.61 |
| ShuffleNet_V2_X2_0 | Image | 30.1 | 15047 | 2.05e-02 | 44.11 | 44.07 | -0.04 |
| SqueezeNet1_0 | Image | 30.7 | 15326 | 2.05e-02 | 23.98 | 24.02 | 0.04 |
| Swin_B | Image | 6.3 | 3157 | 2.05e-02 | 36.46 | 36.55 | 0.1 |
| Swin_T | Image | 10.1 | 5073 | 2.05e-02 | 35.5 | 35.27 | -0.24 |
| Swin_V2_B | Image | 6.2 | 3096 | 2.05e-02 | 35.56 | 35.82 | 0.26 |
| Swin_V2_T | Image | 9.6 | 4815 | 2.05e-02 | 35.6 | 35.53 | -0.06 |
| VGG11_BN | Image | 18.1 | 9068 | 2.05e-02 | 28.75 | 28.95 | 0.2 |
| VGG19_BN | Image | 13.5 | 6771 | 2.05e-02 | 29.85 | 29.88 | 0.03 |
| ViT_B_16 | Image | 9.0 | 4475 | 2.05e-02 | 34.61 | 34.59 | -0.02 |
| ViT_L_32 | Image | 10.4 | 5176 | 2.05e-02 | 29.37 | 29.73 | 0.37 |
| Wide_ResNet101_2 | Image | 8.9 | 4465 | 2.05e-02 | 31.09 | 33.3 | 2.22 |
| Wide_ResNet50_2 | Image | 9.3 | 4626 | 2.05e-02 | 31.5 | 33.96 | 2.46 |

Table 4: Performance with the same hyperparameters across all configurations $O_{Img}$ and per LLM (i.e, text model)

an increase before), but is shifted up. Despite some investigation, we have not determined why. Thus, arguably one might exclude WideResNet and Alexnet outcomes. The implications of doing so, are primarily on the mean $\Delta_{Acc}$, which might drop by about $(2 \cdot 2.2\%/38) \approx 0.0011 \approx 0.1\%$. Significance scores are not impacted a lot, as we removed just 2 positive configurations with $\Delta_{Acc}$ and we also removed one negative (for AlexNet). Gemma-3 shows large gains. Gemma-3 has a larger vocabulary ,i.e., more than 260k tokens, compared to others all having at most 150k (We did not check Fox-1). We leave architecture specific benchmarks and detailed analysis for future work.

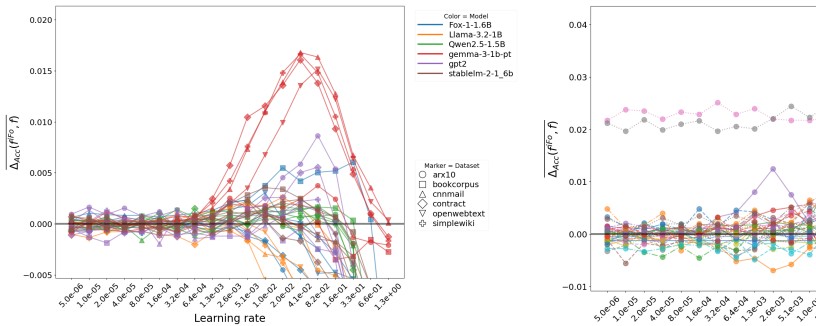

Figure 11: $\Delta_{Acc}$ for different learning rates of all configurations $O_{Text}$

Figure 12: $\Delta_{Acc}$ for different learning rates of all configurations $O_{Img}$

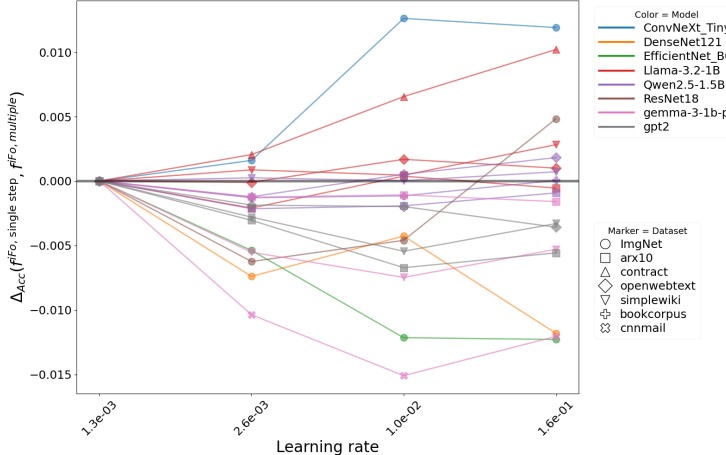

Figure 13: Relative change in predictions $\Delta_{Acc}$ over iterations compared to single step with learning rate $2^{Iter}$ for iFo

### A.3.2    EXPERIMENT 5 - COMPARISON WITH DOMAIN ADAPTATION METHODS

*Hyperparameters and Implementation details:* For SAR and Tent, we relied on defaults from the ReCAP Repo https://github.com/hzcar/ReCAP, which uses a batch size of 1 (like us) but does continual learning (we reset the model after each batch). For PASLE, we used PASLE's own repo https://github.com/palm-ml/PASLE. For the temperature hyperparameter we tried $\{3, 1, 0.8, 0.5\}$ and selected the best outcome based on 10 models, i.e., 0.8. For the (uncertainty) threshold $\tau(r)$, the paper's considered range is $[0.55, 0.95]$. We found that large thresholds like 0.9 gave basically no uncertain samples. We ended up with 0.6, as this gave a similar number of uncertain samples as our method, e.g., on average iFo has about 13% of all samples being uncertain, while PASLE had about 10%. Note that this should be in PASLE's favor, as it is more risky to change less uncertain samples, i.e., iFo gave higher $\Delta_{Acc}$ using a lower percentage of uncertain samples. For learning rates, we chose the best one by looking at rates differing by a factor of 2 (as for our method). More results for other learning rates (with lower average mean) are in Table 5.

*Results: Wall-clock time:* If a sample is optimized Tent, PASLE and iFo all perform two forward and one backward pass plus resetting the model, while SAR requires an extra forward and backward

pass. Other computations are negligible. Thus, all are expected to require very similar time for a sample being optimized. However, iFo is the only method that does not compute on all samples. In turn, in the given setup, where only about 10% of all samples are being optimized by iFo, it is more than a factor of 2 faster. If we increase the ratio of optimized samples say to about 50%, we still expect more than 25% time savings. Under the assumption that at least half of all samples are easy to classify (e.g., exhibit low uncertainty), doing so is not affecting the number of altered samples.

Table 5: Extended Comparison to other methods with more learning rates. Some methods also show gains but all have means worse than our method $iFo$ on ImageNet on $O_{Img}$.

| Method | lr | $\overline{\Delta_{Acc}}$ | Std $\Delta_{Acc}$ | pval $\overline{\Delta_{Acc}^{iFo}} > \overline{\Delta_{Acc}}$ | pval $\overline{\Delta_{Acc}} > 0$ |
|---|---|---|---|---|---|
| SAR | 3.1e-06 | -0.0011 | 0.0109 | 0.0019 | 0.73 |
| SAR | 6.3e-06 | -0.0018 | 0.0120 | 0.0019 | 0.81 |
| SAR | 1.3e-05 | -0.0054 | 0.0197 | 0.0019 | 0.94 |
| SAR | 2.5e-05 | -0.0191 | 0.0488 | 0.0019 | 0.99 |
| SAR | 1.0e-04 | -0.1398 | 0.2680 | 0.0018 | 1.00 |
| SAR | 2.0e-04 | -0.2612 | 0.4009 | 0.0018 | 1.00 |
| Tent | 1.3e-05 | 0.0080 | 0.0145 | 0.0019 | 0.00 |
| Tent | 2.5e-05 | 0.0132 | 0.0235 | 0.0019 | 0.00 |
| Tent | 1.0e-04 | 0.0178 | 0.0445 | 0.0019 | 0.01 |
| Tent | 2.0e-04 | -0.0009 | 0.1272 | 0.0019 | 0.52 |
| Tent | 4.0e-04 | -0.0382 | 0.3091 | 0.0019 | 0.77 |
| PASLE | 2.5e-05 | 0.06 | 0.39 | 0.0019 | 0.16 |
| PASLE | 5.0e-05 | 0.13 | 0.77 | 0.0020 | 0.16 |
| PASLE | 1.0e-04 | -0.03 | 1.17 | 0.0019 | 0.57 |
| PASLE | 2.0e-04 | -0.46 | 3.71 | 0.0017 | 0.77 |
| **Ours, iFo** | 2.05e-02 | 0.28 | 0.55 | - | 0.004 |

### A.3.3 HYPERPARAMETER: EXPERIMENT 6 - ITERATIONS $T$ (SINGLE VS. MANY)

Our algorithm 1 relies on the learning rate $\eta$ and the number of optimization iterations $T$. For computational reasons we would like to minimize iterations and we might even hope for better outcomes if doing so. Thus, we investigate if performing a single step with a large learning rate, leads to similar gains $\Delta_{Acc}$ as conducting many steps with a small learning rate. For multiple iterations, we chose a learning rate of $\eta = 5e-6 \cdot 2^{10} = 0.00512$ and conduct $T = 8$ iterations. Lower learning rates hardly yield gains for a single step, while larger learning rates after $T = 8$ iterations) tend not to give improvements. We compare the multi-step performance against doing just a single step with learning rate $\eta' = \eta \cdot 2^{Power}$ for $Power \in \{0, 1, 2, 3\}$. As measure, we use the difference in accuracy between single and multi step. Results in Figure 13 indicate that single step optimization and multi-step optimization vary between model-dataset pairs but are on average mostly on par. Thus, given the much lower computational overhead for single-step optimization, it seems preferable.

### A.3.4 COMPARISON TO OTHER FINE-TUNING METHODS: EXPERIMENT 7 - COMPARISON TO FINE-TUNING ON INPUTS

We also apply the same fine-tuning mechanism on inputs only denoted as $f^{\text{Tuned on Inputs}}$. For each sample $X$ we perform one fine-tuning step with different learning rates on the given input $X$.

Figures 14 and 15 show outcomes when fine-tuning only on inputs $f^{\text{Tuned on Inputs}}$ in aggregate form as well as using individual model-dataset pairs. They are qualitatively similar to those of fine-tuning on focus classes, i.e., for $f^{iFo}$. It is more interesting to directly compare accuracy gains of both methods $(acc_{f^{iFo}} - acc_f) - (acc_{f^{\text{Tuned on Inputs}}} - acc_f) = acc_{f^{iFo}} - (acc_{f^{\text{Tuned on Inputs}}})$ across learning rates. Figure 17 shows that the balance is both positive and negative for our method $iFo$ compared to fine-tuning on inputs. However, the figure might be a bit misleading as for high learning rates both fine-tuning methods do not perform well and it is not reasonable to apply either of them if it yields no gains. Thus, Figure 17 shows the accuracy gaps, when setting the minimum accuracy difference to 0 for each method, indicating that in such a case it is of no value (and would not be used):

$$\Delta_{Acc,\text{clip acc } 0} := \max(0, (acc_{f^{Opt}} - acc_f)) - \max(0, (acc_{f^{\text{Tuned on Inputs}}} - acc_f))$$

Figure 17 shows that neither method consistently yields better outcomes across all text configurations. For some classifiers like QWEN 2.5, GPT2 and StableLM-2 our method iFo outperforms fine-tuning on inputs on all but one dataset, while for Gemma-3, LLama 3.2 and Fox-1 the opposite holds. This suggests that neither method is a direct replacement of the other and there seem to be situations

where each is preferable. Also note that fine-tuning on inputs (without any labels) is only done for self-supervised learning and it is not easily applicable for models trained in a supervised manner, which holds for the majority of the vision models.

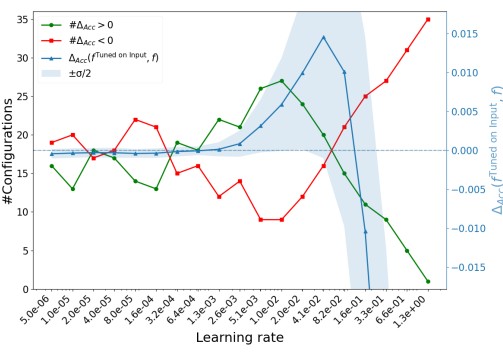

Figure 14: Tuning only on given input $f^{\text{Tuned on Inputs}}$: Average $\Delta_{Acc}$ and $\#\Delta_{Acc} > 0$ for different learning rates for all configurations $O_{Text}$

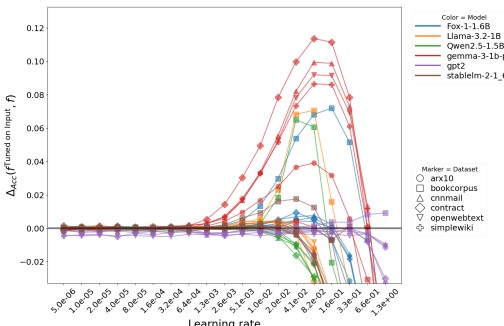

Figure 15: Tuning only on given input $f^{\text{Tuned on Inputs}}$: $\Delta_{Acc}$ for individual text configurations

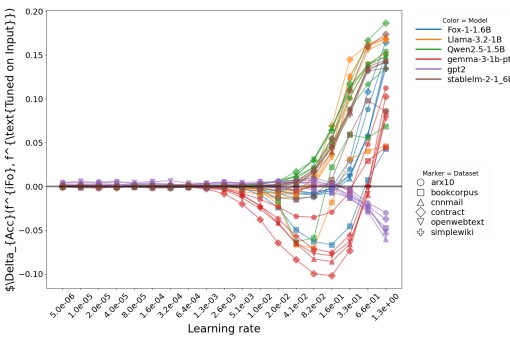

Figure 16: $\Delta_{Acc}$ for our method iFo $f^{iFo}$ and fine-tuning on inputs $f^{\text{Tuned on Inputs}}$ for different learning rates for all configurations $O_{Text}$

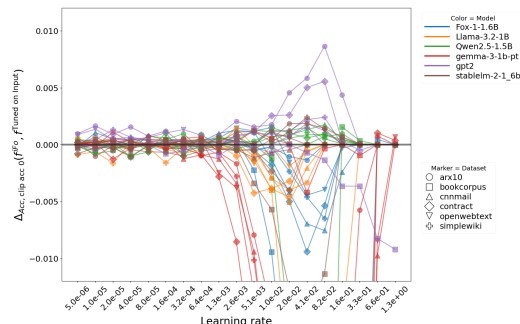

Figure 17: $\Delta_{Acc,\text{clip acc } 0}$, when clipping the minimum accuracy $acc_f$ at 0 for both fine-tuning methods, i.e., iFo $f^{iFo}$ and fine-tuning on inputs $f^{\text{Tuned on Inputs}}$, for different learning rates for all configurations $O_{Text}$.

### A.3.5 ABLATION STUDY: EXPERIMENT 9 - LOSS FUNCTION: CROSS-ENTROPY VS. LOGITS

We suggest to optimize logits in Eq. 2 rather than the more prevalent cross-entropy loss. The reasoning being that cross-entropy would introduce a conceptual mix of our ideas embedded in doFo or iFo (as explained in the Appendix A.1). However, it is not clear whether from a poor empirical statement logits are beneficial. Our results (Table 6) show that using logits as loss is better on text (mean accuracy larger with p-value 0.02) but no significant differences for images.

### A.3.6 ABLATION STUDY: EXPERIMENT 10 - LOSS FUNCTION: ENTROPY VS. LOGITS

We compare optimizing for minimal entropy rather than logits as suggested in Eq. 2, which is common in many domain-adaptation works (see e.g. Table 3). Results for entropy (Table 7) yield no statistical differences using mean accuracy. Comparing the number of correct samples of models $\sum_{o \in O} |D_{te,1,2}| \cdot \Delta_{Acc}$ for $O \in \{O_{Text}, O_{Img}\}$ rather than model accuracy (which gives more weight to models with many uncertain samples) yields that logits are better (p-value 0.02) but no statistical significant differences on images.

### A.3.7 Additional experiment: Experiment 11 - Using instruction-tuned models on Multiple Choice Benchmarks

We also used instruction tuned models and standard benchmarks from HuggingFace, i.e., MMLU (cais/mmlu), ARC (ai2_arc), OpenBookQA (openbookqa), TruthfulQA (EleutherAI/truthful_qa_mc), WinoGrande (allenai/winogrande), HellaSwag (Rowan/hellaswag), CommonsenseQA (commonsense_qa) and restricted outputs to choices, e.g., $A$, $B$, $C$ and $D$ for four choices. We tuned on the two most likely outcome of the choices for iFo (e.g., tokens $A$, $B$,...) rather than on the two most likely tokens. Outcomes of standard multiple choice benchmarks depicted in Table 8 suggest benefits of iFo, when choosing an optimal learning rate per classifier. But the learning rates are significantly wider spread than for non-instruction tuned models, e.g., they range from about 5e-5 to 0.5 (compared to about 2e-3 to 4e-2), suggesting unstable results. The aggregate across all model-dataset pairs (Figure 18) looks rather noisy compared to non-instruction-tuned models and statistical tests show no significant gains. Figures 19 shows individual configurations. The reason for the noisy behavior could be manifold, e.g., one should tune on the two most likely tokens (rather than the two most likely choices $A - X$). Note: For gemma only few samples exhibit high uncertainty for $d_{1,2} = 0.16$, i.e., as few as five samples. Thus, we used $d_{1,2} = 0.84$.

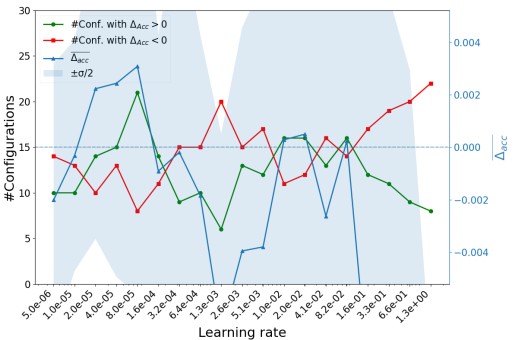

Figure 18: Multiple choice benchmarks, tuning for most likely choices $A - X$: Average $\Delta_{Acc}$ and $\#\Delta_{Acc} > 0$ for different learning rates for all configurations $O_{Text}$

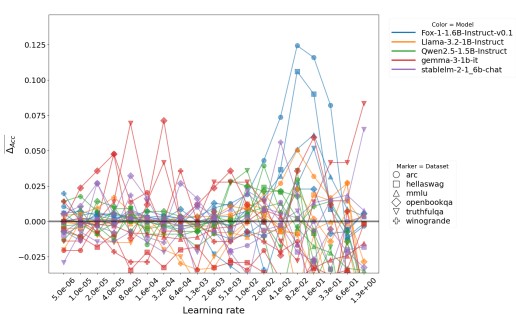

Figure 19: Multiple choice benchmarks, tuning for most likely choices $A - X$: $\Delta_{Acc}$ for individual text configurations

Details on the prompting: We used the following prompt for our multiple choice benchmarks enhanced with model specific templates for instructions:
*"Answer the multiple-choice question. State only the letter ({letters}). Question: {question}*
*n Choices: {options}*
*n Answer:*
*n"* Letters are given by the number of choices (e.g., for four choice questions letters="A, B, C, or D").

We did not parse for the response but picked the token corresponding to the letter with highest output as prediction, irrespective of whether another token had a larger output. This strategy is common and avoids a potentially unreliable parsing process, while being extremely token efficient, i.e., we only need one output token.

### A.4 Extra plots

Figure 20 shows the same pattern as for the mean for each individual sample without any noticeable number of exceptions.

Figure 21 shows the relation between fraction of uncertain samples $\frac{|D_{te,1,2}|}{|D|}$ and gain in terms in accuracy $\Delta_{Acc}$ for individual configurations. It indicates that that while for some configurations the decrease of $\Delta_{Acc}$ with $d_{1,2}$ is rapid for others it is much less also indicated by the large standard deviation in the aggregate plot (Figure 6).

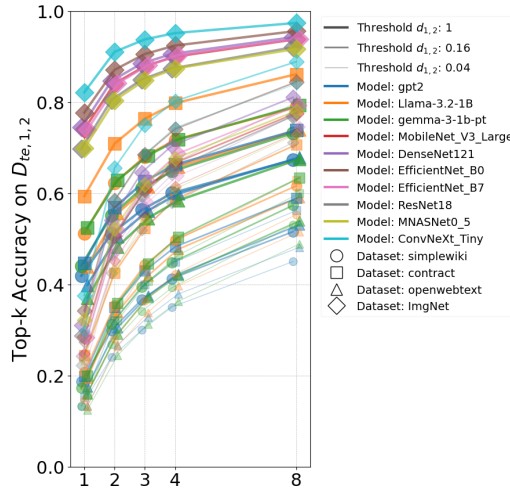

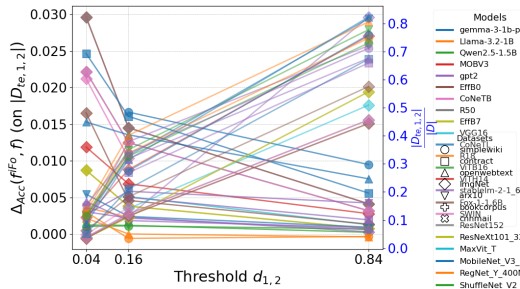

Figure 21: For individual configurations: Larger threshold $d_{1,2}$ implies more samples are uncertain ($\frac{|D_{te,1,2}|}{|D|}$ grows) and optimized using our methods, but the relative gain in terms of accuracy $\Delta_{Acc}$ on uncertain samples $D_{te,1,2}$ declines.

Figure 20: For individual configurations: Difference in Top-k Accuracy for non-tuned models $f$ for $k = 1$ and $k = 2$ grows with lower threshold $d_{1,2}$.

## B  FULL THEORETICAL MOTIVATION

We provide added text compared to shortened motivation in Section 3 using *italic*.

We provide intuition by relating our method—optimizing multiple focus classes $F$—to the scenario of optimizing toward a single class, as is common in typical cross-entropy loss computations. Using a simple model and gradient descent equations, we examine which features become more relevant, distinguishing between those shared among classes and those unique to specific classes. We also discuss why a single-step optimization with a larger learning rate can lead to rapid amplification of shared features and be less stable (for iFo) compared to multi-step optimization with a smaller learning rate.

**Model:**  We assume three output classes $C = \{y_0, y_1, y_2\}$ and focus classes $F = \{y_0, y_1\}$. We have four input features $X = \{x_0, x_1, x_2, x_3\}$. The input features $x_j$ can be seen as originating from a prior layer $l$ with parameters $c'$ processing activations $z$, i.e., $x_j = l(z|c')$. Outputs $y_i := f_i(X)$ are computed as:

$$y_0 = c_0 x_0 + c_4 x_3; \qquad y_1 = c_1 x_1 + c_5 x_3; \qquad y_2 = c_2 x_2 + c_6 x_3 \tag{5}$$

We apply the intuitive notion that a feature value cannot be negative, i.e., $x_i \geq 0$, which happens, for example, after layer activations pass through a non-linearity like $ReLU$. The model implicitly expresses that the presence of input features $\{x_0, x_1, x_2\}$ is only indicative of class $i$, i.e., a change of $x_i$ alters only $y_i$ for $i \in \{0, 1, 2\}$. In turn, $c_0 > 0$, $c_1 > 0$, and $c_2 > 0$. The shared feature $x_3$ impacts all outputs $y_i$. It might also be contrastive, i.e., it can be that two parameters from $\{c_4, c_5, c_6\}$ have opposing signs. Depending on the sign of $c_4, c_5$ and $c_6$ the presence of the feature ($x_3 > 0$) increases or decreases $y_i$. As we model dependence using shared parameters and activations through $x_3$, we assume that $x_j$ are independent, i.e., rely on different parameters and activations. This also facilitates our analysis.

*We consider updates to parameters $c_i$ depending on the loss from our methods iFo $L^{iFo} = -(y_0 + y_1)$ (Eq. 2 without weighting), doFo $L^{doFo} = y_2$ (Eq. 2) and from optimizing just a single class either increasing $L_{y_i,+} = y_i$ or decreasing it $L_{y_i,-} = -y_i$. In short, $L_{y_i,\pm} = \pm y_i$. Note, that for maximizing a class output $y_i$, the loss is $L_{y_i,-} = -y_i$. We have that $L^{iFo} = -(y_0 + y_1) = L_{y_0,-} + L_{y_1,-}$ and, analogously, $L^{doFo} = y_2 = L_{y_2,+}$.*

*We investigate updates to parameters $c_i$ due to backpropagation:*

$$c_i \leftarrow c_i - \eta \frac{\partial L}{\partial c_i} \tag{6}$$

*with $L \in \{L^{iFo}, L^{doFo}, L_{y_i, \pm}\}$*

**Analysis:** *Let us compute updates to $c_i$. We need the following partial derivatives:*

$$\frac{\partial L_{y_j, \pm}}{\partial c_i} = \begin{cases} \pm x_j, & \text{if } i = j \\ \pm x_3, & \text{if } i = 4 + j \\ 0, & \text{otherwise} \end{cases} \tag{7}$$

$$\frac{\partial L^{iFo}}{\partial c_i} = \begin{cases} -x_j, & \text{if } i = j, j \in \{0, 1\} \\ -x_3, & \text{if } i = j, j \in \{4, 5\} \\ 0, & \text{otherwise} \end{cases} \tag{8}$$

$$\frac{\partial L^{doFo}}{\partial c_i} = \begin{cases} x_2, & \text{if } i = 2 \\ x_3, & \text{if } i = 6 \\ 0, & \text{otherwise} \end{cases} \tag{9}$$

Partial derivatives of the loss with respect to $z_i$:

$$\frac{\partial L_{y_j, \pm}}{\partial z_i} = \pm \left( c_j \frac{\partial x_j}{\partial z_i} + c_{4+j} \frac{\partial x_3}{\partial z_i} \right) \tag{10}$$

$$\frac{\partial L^{iFo}}{\partial z_i} = -c_0 \frac{\partial x_0}{\partial z_i} - c_1 \frac{\partial x_1}{\partial z_i} - (c_4 + c_5) \frac{\partial x_3}{\partial z_i} \tag{11}$$

$$\frac{\partial L^{doFo}}{\partial z_i} = c_2 \frac{\partial x_2}{\partial z_i} + c_6 \frac{\partial x_3}{\partial z_i} \tag{12}$$

Let us compare the updates for loss $L^{iFo}$ (without weighting) using both focus classes $F = \{y_0, y_1\}$ and using just either $y_0$ or $y_1$, i.e., the loss $L_{y_0, -}$ or $L_{y_1, -}$. Wlog., we use $L_{y_0, -}$.

The changes related to parameters $\{c_0, c_1\}$ tied to a class-specific feature are identical, e.g., $\frac{\partial L^{iFo}}{\partial c_0} = \frac{\partial L_{y_0, -}}{\partial c_0}$. However, the behavior for the shared feature $x_3$ differs between iFo and single class optimization. The feature's impact on outputs can grow or diminish disproportionally: Say both $c_4$ and $c_5$ have the same sign. Assume $c_4 > 0$ and $c_5 > 0$. Then $\frac{\partial L^{iFo}}{\partial z_i} > \frac{\partial L_{y_0, -}}{\partial z_i}$ as $c_4 + c_5 > c_4$. Thus, the change to the shared feature $x_3$ is larger for $L^{iFo}$. Thus, in the next update also $c_4$ (and $c_5$) are changed more strongly than for single class optimization as $\frac{\partial L^{iFo}}{\partial c_i} = x_3$ for $i \in \{4, 5\}$. Thus, there is a "double growth effect," where the growth of $x_3$ amplifies the growth of parameters $c_4$ and $c_5$ and vice-versa. This could lead to instability in an iterative process, e.g., the shared feature becomes the sole decision criterion with exploding coefficients. This "double effect" is not present if we perform just a single step (with large learning rate). However, for both single and multi-step optimization we observe that the shared feature is altered more. If both $c_4$ and $c_5$ have different signs, i.e., the changes to $x_3$ are less for iFo than for optimizing only $y_0$ and in turn also $c_4$ and $c_5$ change less. Put differently, the relevance of $x_3$ to the decision process diminishes relative to single class optimization.

*Intuitively, if $c_4 \approx c_5$ (both coefficients have the same sign and the same magnitude), i.e., $x_3$ contributes positively to $y_0$ and $y_1$ then feature $x_3$ becomes even more relevant compared to other features. In turn, the role of non-shared features diminishes. This includes features $x_0$ and $x_1$ that are only specific to each focus class $y_0$ and $y_1$. It also covers features that are shared with other classes and just one focus class.[1] On the one hand relying more on shared features seems flawed as class-specific features that matter only for one of the (focus) classes also play a role in discriminating them. However, the fact that we have high uncertainty and we know that the classifier deemed both classes $y_0$ and $y_1$ relevant, justifies that also the shared features play a bigger role as their activation are either relatively large compared to any of the class-specific feature or the class-specific features are of similar magnitude. One might exclude this case by using class-specific features more explicitly to discriminate, e.g., $y_0 = c_0 x_0 + c_4 x_3 - c_7 x_1 - c_8 x_2$, which we did not do for readability.*

*Let us compare the updates for loss $L^{doFo}$ with $L_{y_0, -}$. Say $c_2$ and $c_3$ have the same sign, i.e., feature $x_3$ increases outputs for all classes or decreases it. Then using $L_{y_0, -}$ increases the relevance of all*

---

[1]which we did not include in our model for simplicity, but we might as well assume that there exists a $y_3$ or even $y_2$ that also depends on $x_0$, which would not change any of our computations for iFo and classes $y_0, y_1, y_2$.

*features $c_0$ and $c_4$, while doFo decreases $c_2$ and the shared feature $x_3$, leading to a stronger reliance of $y_0$ on features only present $y_0$, i.e., $x_0$. This behavior is different from iFo, as for iFo $x_3$ would become more relevant. For, doFo features of out-of-focus classes are deemed irrelevant. Say $c_4 > 0$ and $c_5 < 0$ (or vice versa), e.g., $c_4 \approx -c_5$. As $x_3$ decreases for doFo, $y_1$ grows, while $y_0$ shrinks. This behavior is also different from iFo, since if $c_4 \approx -c_5$ there is no net impact on $c_3$.*

In summary, ***iFo amplifies features that are shared between focus classes*** $F$, while doFo hampers features that are shared between focus and out-of-focus classes.

**Relation to entropy minimization:**  Let us also compare to entropy minimization with loss $L^H = H = -\sum_k p_k \log p_k$. Let us assume that $y_0 = y_1 \geq y_2 \geq 0$ and in turn $p_0 = p_1 \geq p_2$. This is the most important case, as it mimics the situation that the first two classes are very likely but exhibit high uncertainty and the others potentially much less. Thus, changing the outputs comes with smallest possible risk of errors and requires only small updates to the network. Furthermore, if the most likely class is much larger than the second most likely, i.e., in our case $y_0 \gg y_1$, optimization has generally no impact and we do not attempt it.

Partial derivatives with respect to logits $y_i$:

$$g_k := \frac{\partial H}{\partial y_k} = p_k\big(-H - \log p_k\big).$$

With the symmetry $p_0 = p_1$, we have

$$g_0 = g_1 =: g_a^{minH} < 0, \qquad g_2 =: g_b^{minH} > 0,$$

Partial derivatives with respect to $c_i$:

$$\frac{\partial L^H}{\partial c_0} = g_a^{minH} x_0, \quad \frac{\partial L^H}{\partial c_1} = g_a^{minH} x_1, \quad \frac{\partial L^H}{\partial c_2} = g_b^{minH} x_2,$$

$$\frac{\partial L^H}{\partial c_4} = g_a^{minH} x_3, \quad \frac{\partial L^H}{\partial c_5} = g_a^{minH} x_3, \quad \frac{\partial L^H}{\partial c_6} = g_b^{minH} x_3$$

Thus, $c_0, c_1, c_4, c_5$ increase since $g_a^{minH} < 0$. $c_2, c_6$ decrease since $g_b^{minH} > 0$.

The derivatives for entropy minimization $\frac{\partial L^H}{\partial c_i}$ equal those of iFo $\frac{\partial L^{iFo}}{\partial c_i}$ except for the multiplication with $g^{minH}$, which are fixed to 1 for iFo – we introduce the coefficient $\alpha^{iFo} = 1$ for naming purposes. Note that the gradients are multiplied by the learning rate $\eta$, which can be chosen freely, meaning that the coefficients $g, \alpha$ are scaled by $\eta$. Figure 3 visualizes the coefficients $\alpha^{iFo}, g_a^{\min H}$ and $g_b^{\min H}$. More interestingly, is the non-linearity of $g$. In particular, the coefficients are 0 near $p \approx 0.5$ and $p \approx 1/3$. This means that in this case the prediction will not be changed, no matter what the learning rate $\eta$ is. However, following our rationale, in both we have lowest risks when making changes to the prediction. That is, for $p \approx 0.5$ the uncertainty among the top classes is largest and there are no alternative options. For $p \approx 1/3$ the uncertainty among the top three classes is largest as all have the same probability $p_0 = p_1 = p_2 = 1/3$ and choosing any of them is reasonable. Thus, entropy is poorly aligned with our rationale. Using constant coefficients $\alpha_a^{iFo}$ is better suited.

*For the upstream variable $z_i$ producing the features,*

$$\frac{\partial L^H}{\partial z_i} = \sum_{j=0}^{2} g_j \bigg( c_j \frac{\partial x_j}{\partial z_i} + c_{4+j} \frac{\partial x_3}{\partial z_i} \bigg) = g_a^{minH} \bigg( c_0 \frac{\partial x_0}{\partial z_i} + c_1 \frac{\partial x_1}{\partial z_i} + (c_4 + c_5) \frac{\partial x_3}{\partial z_i} \bigg) + g_b^{minH} \bigg( c_2 \frac{\partial x_2}{\partial z_i} + c_6 \frac{\partial x_3}{\partial z_i} \bigg).$$

*Because $g_a^{minH} < 0 < g_b^{minH}$, the shared-feature term combines as*

$$\big[g_a^{minH}(c_4 + c_5) + g_b^{minH} c_6\big] \frac{\partial x_3}{\partial z_i}. \big[g_a^{\min H}(c_4 + c_5) + g_b^{\min H} c_6\big] \frac{\partial x_3}{\partial z_i}.$$

*Thus, entropy minimization mirrors a reduced "double growth" mechanism compared to iFo: the shared path gets reinforced for the currently favored classes and damped for the unfavored one, which in turn further separates the logits. But in contrast to $L^{iFo}$ we note that the impact on shared features is lower because $g_a^{minH}$ and $g_b^{minH}$ have opposite signs. Though as $|g_b^{\min H}| \leq |g_a^{\min H}|$ the overall effect remains intact.*

## B.1 RESULTS FOR DOFO

We evaluated doFo only on a subset of image and text configurations (see Figures 24 and 25) as it became evident that it yielded gains only in isolated cases using $|F \setminus C| \in \{|C|-2, |C|-6, |C|-18\}$. We show only those for $|C| - 2$ in the referenced figures. The aggregate outcomes in Figures 22 and 23 show no benefits for doFo. There are a few exceptions like Gemma-3 that indicate rather favorable outcomes. As elaborated on in the discussion there can be multiple reasons, why results are not favorable, some of which could be explored more in future research.

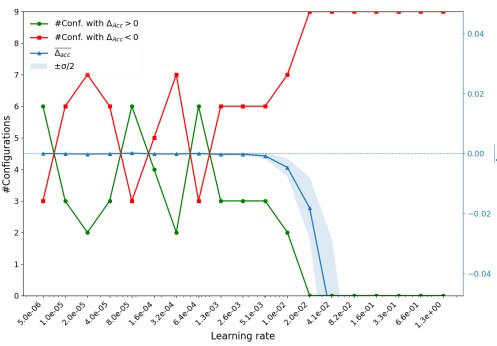
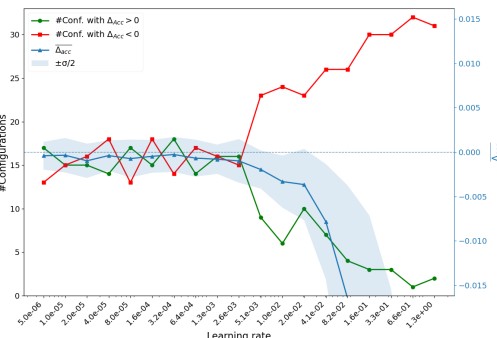

Figure 22: Method doFo: Average $\Delta_{Acc}$ and $\#\Delta_{Acc} > 0$ for different learning rates for all configurations $O_{Text}$

Figure 23: Method doFo: Average $\Delta_{Acc}$ and $\#\Delta_{Acc} > 0$ for different learning rates for all configurations $O_{Img}$

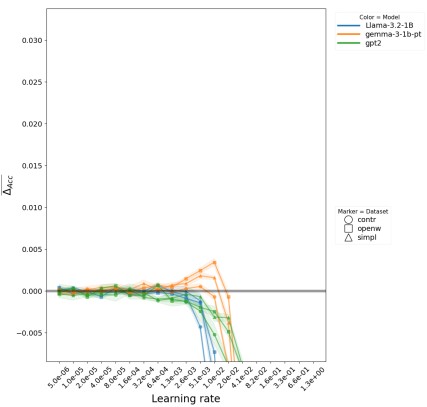
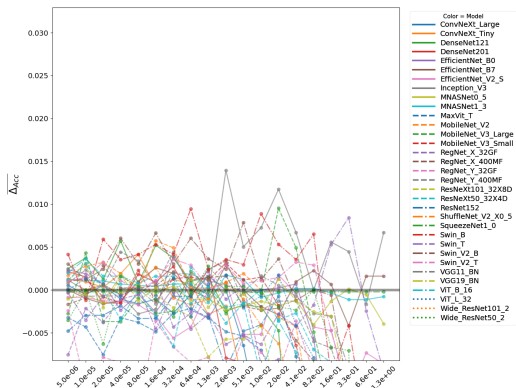

Figure 24: Method doFo: $\Delta_{Acc}$ for different learning rates for a subset of configurations $O_{Text}$

Figure 25: Method doFo: $\Delta_{Acc}$ for different learning rates configurations $O_{Img}$

## B.2 ESSENTIAL CODE

Below we put the most essential code for our methods $iFo$ and $doFo$. More code is in the supplement.

Listing 1: Training/tuning loop

```
cfg={'learning_rate': 5e-6,  #initial learning rate
"backIter": 19, #learning rate increments
"It1Inc": 2, #if 0 do classicl multistep fine-tuning, if >0 single step (
    by default do single-step)
"gThres": (0, 0.16), # 0.16 is threshold d_(1,2)
"seed": 0, #random seed
"incTop": (1, 2), #2...number of focus classes |F|
"maxSamp": 20000,
"wei": (2,), #weighting  (if (0,) then don't), default use weighting
"weight_decay":0,'beta1':0,'beta2':0,'grClip':1, #gradient clipping
"fTuOther":0 #fine-tune on inputs first (not done by default)
```

```python
}

model = ... #get from huggingface
dataset = ... #(x,y) - tokenized data from huggingface or other sources
    after filtering for uncertainty d_(1,2)

with ctx:
    if not mustClone: state = model.state_dict()
    clonecl = copy.deepcopy(model)

logger = regutilsNew.RegLogger(cfg, cres)
if not isVision: optimPara = getOptPara(clonecl, cfg["weight_decay"]) if
    not mustClone else None
if cfg["It1Inc"] > 0:
    totlrs = [cfg["learning_rate"] * (cfg["It1Inc"] ** (j - 1)) for j in
        range(cfg["backIter"] + 1)]
    testIters = list(range(1,cfg["backIter"]+1))
#printStart,pval = 0,-1
for iter_num in range(okx.shape[0]):
    x, y = okx[iter_num].unsqueeze(0), oky[iter_num].unsqueeze(0)
    if mustClone:
        del clonecl
        gc.collect()
        torch.cuda.empty_cache()
        clonecl = copy.deepcopy(model)
    else: clonecl.load_state_dict(state)
    if isVision:
        optimizer = torch.optim.SGD(clonecl.parameters(), lr=cfg["
            learning_rate"], weight_decay=cfg["weight_decay"], momentum
            =0)
    else:
        optimizer = configure_optimizers(clonecl, cfg["weight_decay"],
            cfg["learning_rate"], (cfg["beta1"], cfg["beta2"]),  optType=
            cfg["opt"],optimPara=optimPara)
        if cfg["fTuOther"]: optIn = configure_optimizers(clonecl, cfg["
            weight_decay"], cfg["fTuOther"][1], (cfg["beta1"], cfg["beta2
            "]), optType=cfg["opt"],optimPara=optimPara)
    scaler=torch.cuda.amp.GradScaler(enabled=(dtype == 'float16'))
    dsy = y[0] if isVision else y[0][-1]
    logger.resetSample(dsy)
    if cfg["It1Inc"]>0:
        if not isVision and cfg["fTuOther"]:# tuneInputs only for text
            scaler2 = torch.cuda.amp.GradScaler(enabled=(dtype ==
                'float16'))
            with torch.no_grad(): #get original prediction
                clout = clonecl(x)[0][0, -1] #else: clout =
                    clonecl(x)[0][0,-1]
                idxnext = torch.argmax(clout.detach(), dim=-1)
                coOrg += (idxnext == dsy).item()
            with ctx: #tune on all inputs
                clonecl.zero_grad(set_to_none=True)
                if isGPT2: logits,floss = clonecl(x[:,:-1], y[:,
                    :-1]) #don't use last token to predict
                else:
                    logits= clonecl(x[:, :-1])[0]
                    floss = F.cross_entropy(logits.view(-1,
                        logits.size(-1)), y[:, :-1].view(-1),
                        ignore_index=-1)
                #print(floss,"floss")
                scaler2.scale(floss).backward()  # backward pass,
                    with gradient scaling if training in fp16
                if cfg["grClip"] != 0.0:  # clip the gradient
                    scaler2.unscale_(optIn)
                    torch.nn.utils.clip_grad_norm_(clonecl.parameters
                        (), cfg["grClip"])
```

```
61                    scaler2.step(optIn)
62                    scaler2.update()
63                    with torch.no_grad(): #get prediction after tuning
64                        clout = clonecl(x)[0][0, -1]
65                        idxnext2 = torch.argmax(clout.detach(), dim=-1)
66                        coFt += (idxnext2 == dsy).item()
67                        changedFt= (idxnext2 != idxnext).item()
68
69        with ctx:
70            clonecl.zero_grad(set_to_none=True)
71            if isVision: clout=clonecl(x)[0]
72            else:
73                clout = clonecl(x)[0][0, -1]
74                if isMultiChoice: clout=clout+mask
75            with torch.no_grad(): clo = F.softmax(clout, dim=-1).detach()
76            loss, _ = regutilsNew.getLoss(cfg, clo, clout)
77            logger.updateLRIter(clo, 0)
78
79        scaler.scale(loss).backward()
80        if cfg["grClip"] != 0.0:  # clip the gradient
81            scaler.unscale_(optimizer)
82            torch.nn.utils.clip_grad_norm_(clonecl.parameters(), cfg["
                grClip"])
83        scaler.update()
84
85        params,grads = [],[]
86        for g in optimizer.param_groups:
87            for p in g['params']:
88                if p.grad is not None:
89                    params.append(p)
90                    grads.append(p.grad)
91
92        totlr=0
93        with torch.inference_mode(),ctx:
94          for j in testIters:  # num_repeated_steps is the number of
                times you want to apply the step
95            new_lr = totlrs[j]-totlr # Set your desired learning rate
                here
96            totlr+=new_lr
97            torch._foreach_add_(params, grads, alpha=-new_lr)
98            if isVision: clout = clonecl(x)[0]
99            else:
100                clout = clonecl(x)[0][0, -1]
101                if isMultiChoice: clout = clout + mask
102            clo = F.softmax(clout, dim=-1).detach()
103            logger.updateLRIter(clo, j)
104    if cfg["It1Inc"]==0:
105        for j in range(cfg["backIter"]):
106            with ctx:
107                clonecl.zero_grad(set_to_none=True)
108                if isVision:
109                    clout = clonecl(x)[0]
110                else:
111                    clout = clonecl(x)[0][0, -1]
112                    if isMultiChoice: clout = clout + mask
113                with torch.no_grad():
114                    clo = F.softmax(clout, dim=-1).detach()
115                loss, _ = regutilsNew.getLoss(cfg, clo, clout)
116                if j==0: logger.updateLRIter(clo, j)
117            scaler.scale(loss).backward()
118            if cfg["grClip"] != 0.0:  # clip the gradient
119                scaler.unscale_(optimizer)
120                torch.nn.utils.clip_grad_norm_(clonecl.parameters(), cfg[
                    "grClip"])
121            scaler.step(optimizer)
```

```
122          scaler.update()
123          optimizer.zero_grad(set_to_none=True)
124          logger.updateLRIter(clo, j+1)
```

### B.3 LLM USAGE IN WRITING

LLMs, i.e., ChatGPT 5, served as an assistant. They *did not contribute* to any key ideas. They supported in the generation of code for plots, the derivation of the softmax outputs in Section A.1, and polishing the write-up.

Table 6: Performance for iFo with network dependent learning rate (lr) using cross-entropy loss (instead of logits)

| Model | Dataset | $\frac{|D_{te,1,2}|}{|D|}$ | $|D_{te,1,2}|$ | lr | **Acc.**$f$ (Baseline) | **Acc.**$f^{Opt}$ (Ours, doFO) | $\Delta_{Acc}$ |
|---|---|---|---|---|---|---|---|
| Fox-1-1.6B | arx10 | 39.3 | 20000 | 5.12e-03 | 19.41 | 19.5 | 0.09 |
| Fox-1-1.6B | bookc | 64.3 | 20000 | 5.12e-03 | 12.63 | 12.91 | 0.28 |
| Fox-1-1.6B | cnnma | 42.3 | 20000 | 5.12e-03 | 20.57 | 20.7 | 0.13 |
| Fox-1-1.6B | contr | 29.0 | 20000 | 5.12e-03 | 23.42 | 23.27 | -0.15 |
| Fox-1-1.6B | openw | 41.4 | 20000 | 5.12e-03 | 20.3 | 20.63 | 0.33 |
| Fox-1-1.6B | simpl | 36.2 | 20000 | 5.12e-03 | 20.29 | 20.42 | 0.13 |
| Llama-3.2-1B | arx10 | 37.3 | 20000 | 2.56e-03 | 21.47 | 21.54 | 0.07 |
| Llama-3.2-1B | bookc | 61.3 | 20000 | 2.56e-03 | 14.3 | 14.41 | 0.11 |
| Llama-3.2-1B | cnnma | 40.5 | 20000 | 2.56e-03 | 20.8 | 20.82 | 0.02 |
| Llama-3.2-1B | contr | 25.9 | 20000 | 2.56e-03 | 24.45 | 24.68 | 0.23 |
| Llama-3.2-1B | openw | 40.3 | 20000 | 2.56e-03 | 19.78 | 19.79 | 0.01 |
| Llama-3.2-1B | simpl | 33.8 | 20000 | 2.56e-03 | 20.4 | 20.81 | 0.41 |
| Qwen2.5-1.5B | arx10 | 34.9 | 20000 | 1.02e-02 | 20.42 | 20.55 | 0.13 |
| Qwen2.5-1.5B | bookc | 50.7 | 20000 | 1.02e-02 | 15.06 | 15.21 | 0.15 |
| Qwen2.5-1.5B | cnnma | 38.7 | 20000 | 1.02e-02 | 20.76 | 20.83 | 0.07 |
| Qwen2.5-1.5B | contr | 21.5 | 20000 | 1.02e-02 | 26.1 | 26.35 | 0.25 |
| Qwen2.5-1.5B | openw | 40.1 | 20000 | 1.02e-02 | 20.18 | 20.07 | -0.11 |
| Qwen2.5-1.5B | simpl | 33.1 | 20000 | 1.02e-02 | 20.4 | 20.51 | 0.11 |
| gemma-3-1b-pt | arx10 | 33.0 | 20000 | 1.28e-03 | 16.96 | 16.79 | -0.17 |
| gemma-3-1b-pt | bookc | 61.2 | 20000 | 1.28e-03 | 13.02 | 13.03 | 0.01 |
| gemma-3-1b-pt | cnnma | 33.3 | 20000 | 1.28e-03 | 17.01 | 17.22 | 0.21 |
| gemma-3-1b-pt | contr | 21.1 | 20000 | 1.28e-03 | 19.69 | 19.57 | -0.12 |
| gemma-3-1b-pt | openw | 31.9 | 20000 | 1.28e-03 | 16.04 | 16.16 | 0.12 |
| gemma-3-1b-pt | simpl | 27.8 | 20000 | 1.28e-03 | 17.49 | 17.52 | 0.03 |
| gpt2 | arx10 | 37.4 | 20000 | 1.64e-01 | 19.29 | 19.59 | 0.3 |
| gpt2 | bookc | 52.6 | 20000 | 1.64e-01 | 10.59 | 10.48 | -0.11 |
| gpt2 | cnnma | 33.3 | 20000 | 1.64e-01 | 19.55 | 19.51 | -0.04 |
| gpt2 | contr | 26.4 | 20000 | 1.64e-01 | 21.29 | 21.75 | 0.46 |
| gpt2 | openw | 31.8 | 20000 | 1.64e-01 | 19.48 | 19.45 | -0.03 |
| gpt2 | simpl | 34.0 | 20000 | 1.64e-01 | 19.42 | 19.8 | 0.38 |
| stablelm-2-1_6b | arx10 | 38.1 | 20000 | 5.12e-03 | 20.13 | 20.22 | 0.09 |
| stablelm-2-1_6b | bookc | 58.0 | 20000 | 5.12e-03 | 16.6 | 17.2 | 0.6 |
| stablelm-2-1_6b | cnnma | 38.5 | 20000 | 5.12e-03 | 21.27 | 21.36 | 0.09 |
| stablelm-2-1_6b | contr | 19.3 | 20000 | 5.12e-03 | 27.36 | 27.51 | 0.15 |
| stablelm-2-1_6b | openw | 39.3 | 20000 | 5.12e-03 | 20.82 | 20.9 | 0.08 |
| stablelm-2-1_6b | simpl | 31.2 | 20000 | 5.12e-03 | 22.41 | 22.58 | 0.17 |
| AlexNet | Image | 31.3 | 15630 | 5.12e-03 | 23.25 | 23.22 | -0.03 |
| ConvNeXt_Large | Image | 6.9 | 3448 | 5.12e-03 | 37.7 | 37.67 | -0.03 |
| ConvNeXt_Tiny | Image | 11.3 | 5629 | 5.12e-03 | 37.61 | 37.88 | 0.27 |
| DenseNet121 | Image | 14.0 | 7024 | 5.12e-03 | 31.15 | 31.31 | 0.16 |
| DenseNet201 | Image | 11.7 | 5867 | 5.12e-03 | 31.65 | 32.64 | 0.99 |
| EfficientNet_B0 | Image | 14.5 | 7238 | 5.12e-03 | 33.92 | 34.28 | 0.36 |
| EfficientNet_B7 | Image | 11.0 | 5502 | 5.12e-03 | 28.55 | 28.93 | 0.38 |
| EfficientNet_V2_L | Image | 70.9 | 35471 | 5.12e-03 | 2.06 | 2.2 | 0.14 |
| EfficientNet_V2_S | Image | 8.3 | 4162 | 5.12e-03 | 34.19 | 34.5 | 0.31 |
| GoogLeNet | Image | 23.9 | 11957 | 5.12e-03 | 31.45 | 31.54 | 0.09 |
| Inception_V3 | Image | 6.6 | 3318 | 5.12e-03 | 22.63 | 22.66 | 0.03 |
| MNASNet0_5 | Image | 28.0 | 13992 | 5.12e-03 | 30.82 | 30.76 | -0.06 |
| MNASNet1_3 | Image | 36.3 | 18130 | 5.12e-03 | 49.55 | 49.72 | 0.17 |
| MaxVit_T | Image | 6.8 | 3425 | 5.12e-03 | 37.52 | 37.72 | 0.2 |
| MobileNet_V2 | Image | 16.2 | 8096 | 5.12e-03 | 29.21 | 29.41 | 0.2 |
| MobileNet_V3_Large | Image | 11.5 | 5730 | 5.12e-03 | 27.98 | 28.31 | 0.33 |
| MobileNet_V3_Small | Image | 20.0 | 9988 | 5.12e-03 | 25.72 | 26.23 | 0.51 |
| RegNet_X_32GF | Image | 6.7 | 3345 | 5.12e-03 | 31.96 | 32.08 | 0.12 |
| RegNet_X_400MF | Image | 14.5 | 7241 | 5.12e-03 | 30.37 | 30.62 | 0.25 |
| RegNet_Y_32GF | Image | 6.1 | 3061 | 5.12e-03 | 32.38 | 32.9 | 0.52 |
| RegNet_Y_400MF | Image | 13.5 | 6761 | 5.12e-03 | 30.75 | 31.27 | 0.52 |
| ResNeXt101_32X8D | Image | 7.1 | 3529 | 5.12e-03 | 32.76 | 32.19 | -0.57 |
| ResNeXt50_32X4D | Image | 9.2 | 4601 | 5.12e-03 | 32.08 | 32.51 | 0.43 |
| ResNet152 | Image | 9.6 | 4794 | 5.12e-03 | 31.81 | 32.12 | 0.31 |
| ResNet18 | Image | 18.3 | 9128 | 5.12e-03 | 28.6 | 29.06 | 0.46 |
| ShuffleNet_V2_X0_5 | Image | 24.0 | 12014 | 5.12e-03 | 22.32 | 22.59 | 0.27 |
| ShuffleNet_V2_X2_0 | Image | 30.1 | 15050 | 5.12e-03 | 44.1 | 44.32 | 0.22 |
| SqueezeNet1_0 | Image | 30.7 | 15326 | 5.12e-03 | 24.11 | 24.39 | 0.28 |
| Swin_B | Image | 6.3 | 3157 | 5.12e-03 | 36.46 | 36.71 | 0.25 |
| Swin_T | Image | 10.1 | 5073 | 5.12e-03 | 35.5 | 35.48 | -0.02 |
| Swin_V2_B | Image | 6.2 | 3096 | 5.12e-03 | 35.56 | 35.63 | 0.06 |
| Swin_V2_T | Image | 9.6 | 4815 | 5.12e-03 | 35.6 | 35.76 | 0.17 |
| VGG11_BN | Image | 18.1 | 9068 | 5.12e-03 | 28.83 | 28.56 | -0.26 |
| VGG19_BN | Image | 13.5 | 6771 | 5.12e-03 | 29.91 | 30.08 | 0.18 |
| ViT_B_16 | Image | 9.0 | 4475 | 5.12e-03 | 34.64 | 35.06 | 0.42 |
| ViT_L_32 | Image | 10.4 | 5176 | 5.12e-03 | 29.69 | 30.12 | 0.43 |
| Wide_ResNet101_2 | Image | 8.9 | 4467 | 5.12e-03 | 31.05 | 33.67 | 2.62 |
| Wide_ResNet50_2 | Image | 9.3 | 4626 | 5.12e-03 | 31.52 | 33.01 | 1.49 |

Table 7: Performance for iFo with network dependent learning rate (lr) using entropy as loss (instead of logits)

| Model | Dataset | $\frac{|D_{te,1,2}|}{|D|}$ | $|D_{te,1,2}|$ | lr | **Acc.** $f$ (Baseline) | **Acc.** $f^{Opt}$ (Ours, doFO) | $\Delta_{Acc}$ |
|---|---|---|---|---|---|---|---|
| Fox-1-1.6B | arx10 | 39.3 | 10000 | 5.12e-03 | 19.41 | 19.55 | 0.14 |
| Fox-1-1.6B | bookc | 64.3 | 10000 | 5.12e-03 | 12.63 | 12.91 | 0.28 |
| Fox-1-1.6B | cnnma | 42.3 | 10000 | 5.12e-03 | 20.57 | 20.95 | 0.38 |
| Fox-1-1.6B | contr | 29.0 | 10000 | 5.12e-03 | 23.42 | 23.38 | -0.04 |
| Fox-1-1.6B | openw | 41.4 | 10000 | 5.12e-03 | 20.3 | 20.61 | 0.31 |
| Fox-1-1.6B | simpl | 36.2 | 10000 | 5.12e-03 | 20.29 | 20.41 | 0.12 |
| Llama-3.2-1B | arx10 | 37.3 | 10000 | 5.12e-03 | 21.47 | 21.37 | -0.1 |
| Llama-3.2-1B | bookc | 61.3 | 10000 | 5.12e-03 | 14.3 | 14.38 | 0.08 |
| Llama-3.2-1B | cnnma | 40.5 | 10000 | 5.12e-03 | 20.8 | 20.9 | 0.1 |
| Llama-3.2-1B | contr | 25.9 | 10000 | 5.12e-03 | 24.45 | 24.62 | 0.17 |
| Llama-3.2-1B | openw | 40.3 | 10000 | 5.12e-03 | 19.78 | 20.0 | 0.22 |
| Llama-3.2-1B | simpl | 33.8 | 10000 | 5.12e-03 | 20.4 | 20.7 | 0.3 |
| Qwen2.5-1.5B | arx10 | 34.9 | 10000 | 2.05e-02 | 20.42 | 20.44 | 0.02 |
| Qwen2.5-1.5B | bookc | 50.7 | 10000 | 2.05e-02 | 15.06 | 15.17 | 0.11 |
| Qwen2.5-1.5B | cnnma | 38.7 | 10000 | 2.05e-02 | 20.76 | 20.9 | 0.14 |
| Qwen2.5-1.5B | contr | 21.5 | 10000 | 2.05e-02 | 26.1 | 26.31 | 0.21 |
| Qwen2.5-1.5B | openw | 40.1 | 10000 | 2.05e-02 | 20.18 | 20.22 | 0.04 |
| Qwen2.5-1.5B | simpl | 33.1 | 10000 | 2.05e-02 | 20.4 | 20.48 | 0.08 |
| gemma-3-1b-pt | arx10 | 33.0 | 10000 | 4.10e-02 | 16.96 | 16.8 | -0.16 |
| gemma-3-1b-pt | bookc | 61.2 | 10000 | 4.10e-02 | 13.02 | 13.52 | 0.5 |
| gemma-3-1b-pt | cnnma | 33.3 | 10000 | 4.10e-02 | 17.01 | 17.53 | 0.52 |
| gemma-3-1b-pt | contr | 21.1 | 10000 | 4.10e-02 | 19.69 | 20.06 | 0.37 |
| gemma-3-1b-pt | openw | 31.9 | 10000 | 4.10e-02 | 16.04 | 16.54 | 0.5 |
| gemma-3-1b-pt | simpl | 27.8 | 10000 | 4.10e-02 | 17.49 | 18.08 | 0.59 |
| gpt2 | arx10 | 37.4 | 10000 | 1.64e-01 | 19.29 | 19.45 | 0.16 |
| gpt2 | bookc | 52.4 | 100000 | 1.64e-01 | 10.93 | 10.76 | -0.17 |
| gpt2 | cnnma | 33.3 | 10000 | 1.64e-01 | 19.55 | 19.58 | 0.03 |
| gpt2 | contr | 26.4 | 10000 | 1.64e-01 | 21.29 | 21.89 | 0.6 |
| gpt2 | openw | 31.8 | 10000 | 1.64e-01 | 19.48 | 19.4 | -0.08 |
| gpt2 | simpl | 34.0 | 10000 | 1.64e-01 | 19.42 | 19.87 | 0.45 |
| stablelm-2-1_6b | arx10 | 38.1 | 10000 | 4.10e-02 | 20.13 | 20.41 | 0.28 |
| stablelm-2-1_6b | bookc | 58.0 | 10000 | 4.10e-02 | 16.6 | 17.16 | 0.56 |
| stablelm-2-1_6b | cnnma | 38.5 | 10000 | 4.10e-02 | 21.27 | 21.34 | 0.07 |
| stablelm-2-1_6b | contr | 19.3 | 10000 | 4.10e-02 | 27.36 | 27.25 | -0.11 |
| stablelm-2-1_6b | openw | 39.3 | 10000 | 4.10e-02 | 20.82 | 21.05 | 0.23 |
| stablelm-2-1_6b | simpl | 31.2 | 10000 | 4.10e-02 | 22.41 | 22.67 | 0.26 |
| AlexNet | Image | 31.3 | 15630 | 5.12e-03 | 23.25 | 23.21 | -0.04 |
| ConvNeXt_Large | Image | 6.9 | 3448 | 5.12e-03 | 37.7 | 37.3 | -0.41 |
| ConvNeXt_Tiny | Image | 11.3 | 5629 | 5.12e-03 | 37.61 | 37.61 | 0.0 |
| DenseNet121 | Image | 14.0 | 7024 | 5.12e-03 | 31.15 | 31.93 | 0.78 |
| DenseNet201 | Image | 11.7 | 5867 | 5.12e-03 | 31.65 | 32.57 | 0.92 |
| EfficientNet_B0 | Image | 14.5 | 7238 | 5.12e-03 | 33.92 | 34.14 | 0.22 |
| EfficientNet_B7 | Image | 11.0 | 5502 | 5.12e-03 | 28.55 | 28.54 | -0.02 |
| EfficientNet_V2_S | Image | 8.3 | 4162 | 5.12e-03 | 34.19 | 34.48 | 0.29 |
| GoogLeNet | Image | 23.9 | 11957 | 5.12e-03 | 31.45 | 31.43 | -0.02 |
| Inception_V3 | Image | 6.6 | 3318 | 5.12e-03 | 22.63 | 23.06 | 0.42 |
| MNASNet0_5 | Image | 28.0 | 13992 | 5.12e-03 | 30.82 | 31.01 | 0.19 |
| MNASNet1_3 | Image | 36.3 | 18130 | 5.12e-03 | 49.55 | 49.78 | 0.23 |
| MaxVit_T | Image | 6.8 | 3425 | 5.12e-03 | 37.52 | 37.43 | -0.09 |
| MobileNet_V2 | Image | 16.2 | 8096 | 5.12e-03 | 29.21 | 29.63 | 0.42 |
| MobileNet_V3_Large | Image | 11.5 | 5730 | 5.12e-03 | 27.98 | 28.17 | 0.19 |
| MobileNet_V3_Small | Image | 20.0 | 9988 | 5.12e-03 | 25.72 | 26.26 | 0.54 |
| RegNet_X_32GF | Image | 6.7 | 3345 | 5.12e-03 | 31.96 | 32.59 | 0.63 |
| RegNet_X_400MF | Image | 14.5 | 7241 | 5.12e-03 | 30.37 | 30.73 | 0.36 |
| RegNet_Y_32GF | Image | 6.1 | 3061 | 5.12e-03 | 32.38 | 32.83 | 0.46 |
| RegNet_Y_400MF | Image | 13.5 | 6761 | 5.12e-03 | 30.75 | 31.42 | 0.67 |
| ResNeXt101_32X8D | Image | 7.1 | 3529 | 5.12e-03 | 32.76 | 32.9 | 0.14 |
| ResNeXt50_32X4D | Image | 9.2 | 4601 | 5.12e-03 | 32.08 | 32.45 | 0.37 |
| ResNet152 | Image | 9.6 | 4794 | 5.12e-03 | 31.81 | 32.56 | 0.75 |
| ResNet18 | Image | 18.3 | 9128 | 5.12e-03 | 28.6 | 28.98 | 0.37 |
| ShuffleNet_V2_X0_5 | Image | 24.0 | 12014 | 5.12e-03 | 22.32 | 22.99 | 0.67 |
| ShuffleNet_V2_X2_0 | Image | 30.1 | 15050 | 5.12e-03 | 44.1 | 44.16 | 0.06 |
| SqueezeNet1_0 | Image | 30.7 | 15326 | 5.12e-03 | 24.11 | 24.08 | -0.03 |
| Swin_B | Image | 6.3 | 3157 | 5.12e-03 | 36.46 | 36.33 | -0.13 |
| Swin_T | Image | 10.1 | 5073 | 5.12e-03 | 35.5 | 35.44 | -0.06 |
| Swin_V2_B | Image | 6.2 | 3096 | 5.12e-03 | 35.56 | 35.89 | 0.32 |
| Swin_V2_T | Image | 9.6 | 4815 | 5.12e-03 | 35.6 | 35.74 | 0.15 |
| VGG11_BN | Image | 18.1 | 9068 | 5.12e-03 | 28.83 | 29.1 | 0.28 |
| VGG19_BN | Image | 13.5 | 6771 | 5.12e-03 | 29.91 | 29.91 | 0.0 |
| ViT_B_16 | Image | 9.0 | 4475 | 5.12e-03 | 34.64 | 35.04 | 0.4 |
| ViT_L_32 | Image | 10.4 | 5176 | 5.12e-03 | 29.69 | 29.97 | 0.27 |
| Wide_ResNet101_2 | Image | 8.9 | 4467 | 5.12e-03 | 31.05 | 33.6 | 2.55 |
| Wide_ResNet50_2 | Image | 9.3 | 4626 | 5.12e-03 | 31.52 | 32.53 | 1.02 |

Table 8: Performance for iFo with network dependent learning rate (lr) across multiple choice benchmarks $O_{Text}$

| Model | Dataset | $\frac{|D_{te,1,2}|}{|D|}$ | $|D_{te,1,2}|$ | lr | **Acc.**$f$ (Baseline) | **Acc.**$f^{Opt}$ (Ours, doFO) | $\Delta_{Acc}$ |
|---|---|---|---|---|---|---|---|
| Fox-1-1.6B-Instruct-v0.1 | arc | 64.6 | 949 | 8.19e-02 | 24.66 | 37.09 | 12.43 |
| Fox-1-1.6B-Instruct-v0.1 | hella | 53.2 | 2075 | 8.19e-02 | 25.98 | 36.58 | 10.6 |
| Fox-1-1.6B-Instruct-v0.1 | mmlu | 58.1 | 7847 | 8.19e-02 | 24.01 | 29.12 | 5.11 |
| Fox-1-1.6B-Instruct-v0.1 | openb | 69.9 | 699 | 8.19e-02 | 28.33 | 30.62 | 2.29 |
| Fox-1-1.6B-Instruct-v0.1 | truth | 50.9 | 348 | 8.19e-02 | 24.71 | 27.3 | 2.59 |
| Fox-1-1.6B-Instruct-v0.1 | winog | 27.8 | 352 | 8.19e-02 | 49.43 | 27.27 | -22.16 |
| Llama-3.2-1B-Instruct | arc | 40.5 | 595 | 6.55e-01 | 20.84 | 23.53 | 2.69 |
| Llama-3.2-1B-Instruct | hella | 22.7 | 1225 | 6.55e-01 | 25.71 | 27.18 | 1.47 |
| Llama-3.2-1B-Instruct | mmlu | 31.1 | 4314 | 6.55e-01 | 23.83 | 23.67 | -0.16 |
| Llama-3.2-1B-Instruct | openb | 56.1 | 561 | 6.55e-01 | 27.27 | 27.99 | 0.71 |
| Llama-3.2-1B-Instruct | truth | 29.2 | 200 | 6.55e-01 | 32.0 | 30.5 | -1.5 |
| Llama-3.2-1B-Instruct | winog | 74.4 | 941 | 6.55e-01 | 44.53 | 47.29 | 2.76 |
| Qwen2.5-1.5B-Instruct | arc | 20.7 | 304 | 1.02e-02 | 28.62 | 28.62 | 0.0 |
| Qwen2.5-1.5B-Instruct | hella | 17.0 | 951 | 1.02e-02 | 32.39 | 32.81 | 0.42 |
| Qwen2.5-1.5B-Instruct | mmlu | 25.5 | 3526 | 1.02e-02 | 29.38 | 29.98 | 0.6 |
| Qwen2.5-1.5B-Instruct | openb | 28.1 | 281 | 1.02e-02 | 29.89 | 32.38 | 2.49 |
| Qwen2.5-1.5B-Instruct | truth | 16.2 | 111 | 1.02e-02 | 33.33 | 36.94 | 3.6 |
| Qwen2.5-1.5B-Instruct | winog | 22.2 | 281 | 1.02e-02 | 46.26 | 48.75 | 2.49 |
| gemma-3-1b-it | arc | 9.9 | 146 | 4.00e-05 | 23.97 | 28.77 | 4.79 |
| gemma-3-1b-it | hella | 11.6 | 552 | 4.00e-05 | 28.99 | 28.08 | -0.91 |
| gemma-3-1b-it | mmlu | 9.9 | 1360 | 4.00e-05 | 27.28 | 26.99 | -0.29 |
| gemma-3-1b-it | openb | 8.4 | 84 | 4.00e-05 | 25.0 | 29.76 | 4.76 |
| gemma-3-1b-it | truth | 10.5 | 72 | 4.00e-05 | 16.67 | 18.06 | 1.39 |
| gemma-3-1b-it | winog | 11.1 | 140 | 4.00e-05 | 50.0 | 47.86 | -2.14 |
| stablelm-2-1_6b-chat | arc | 20.0 | 294 | 8.00e-05 | 27.55 | 28.23 | 0.68 |
| stablelm-2-1_6b-chat | hella | 25.8 | 1445 | 8.00e-05 | 30.52 | 30.73 | 0.21 |
| stablelm-2-1_6b-chat | mmlu | 20.6 | 2855 | 8.00e-05 | 26.06 | 26.34 | 0.28 |
| stablelm-2-1_6b-chat | openb | 18.5 | 185 | 8.00e-05 | 28.11 | 31.35 | 3.24 |
| stablelm-2-1_6b-chat | truth | 20.2 | 138 | 8.00e-05 | 27.54 | 26.81 | -0.72 |
| stablelm-2-1_6b-chat | winog | 21.2 | 268 | 8.00e-05 | 36.94 | 38.81 | 1.87 |

