# OpenReview forum: "Focus on Likely Classes for Test-Time Prediction"
_ICLR.cc/2026/Conference — Submitted to ICLR 2026_

### Official Review · Reviewer_tKs6 · 2025-10-25

**Soundness:** 2
**Presentation:** 2
**Contribution:** 2
**Rating:** 2
**Confidence:** 3

**Summary:**

The authors propose a test-time fine-tuning approach designed to improve model predictions under uncertainty. Their method encourages the model to focus on the most likely classes in order to refine its outputs. Concretely, they introduce two complementary strategies: doFo (Decrease Out-of-Focus), which suppresses the logits of unlikely classes, and iFo (Increase Focus), which amplifies the logits of likely ones. Extensive experiments across diverse datasets and architectures—covering both text generation and image recognition tasks—demonstrate the effectiveness and generality of the proposed approach.

**Strengths:**

1. The authors conduct extensive experiments across multiple models and datasets on both text and image tasks to validate the effectiveness of doFo and iFo, demonstrating the comprehensiveness of their evaluation.
2. Figure 1 and Figure 2 effectively illustrate the core ideas and workflow of the proposed approach, making the methodology easy to understand for readers.

**Weaknesses:**

1. In the field of test-time adaptation, there exists a related approach called PASLE [1], which partitions data into confident and uncertain subsets—assigning one-hot labels to confident samples and candidate label sets to uncertain ones. Its mechanism of assigning 1 to likely classes and 0 to unlikely ones is conceptually similar to the authors’ strategy of “focusing on the likely classes.” However, this prior work is not discussed or compared in the paper. I recommend the authors include a discussion and experimental comparison with PASLE to clarify the novelty and advantage of their method.
2. The experimental section lacks comparisons with any baseline methods, which makes it difficult to assess the absolute performance of the proposed approach. The authors are encouraged to identify and include several relevant methods from the literature for empirical comparison, especially from adjacent domains such as test-time adaptation.
3. The manuscript contains several overly long paragraphs that affect readability. The authors are advised to split long paragraphs into shorter ones to improve clarity, logical flow, and overall presentation quality.

[1] Selective Label Enhancement Learning for Test-Time Adaptation. ICLR 2025

**Questions:**

See weaknesses.

---

> ### Author Response · Authors · 2025-11-24
> **PASLE learns from (many out-of-domain) samples; we use a single, in-domain sample; thus, relation non-obvious; Method also differ; we outperform**
>
> **Thank you and apologies for not discussing [1] right away.**
> *Similar to PASLE [1]:*
> >- **Differences in problem studied**: **We don't consider domain adaptation like [1] using many samples but improving at test time on a single in-domain sample.** This problem is different from [1] (and other papers). We updated the Abstract/Intro and provided tables so that for experts in TTA/domain adaptation, differences are clear early on.
> >- *Motivation*: Our study is motivated by the assumpation that shared features of likely classes of a single sample are more to be trusted. PASLE's is that given many samples, one should not assign a single (pseudo) label for samples with high uncertainty but only to certain ones. The motivation also has implications on the method:
> >- *Method difference*: Splitting: Our motivation for splitting is to save on unnecessary computation, while PASLE’s motivation is higher accuracy. PASLE optimizes all samples; we do not.
>   We want to decide betwen just two classes that are both very likely. PASLE considers (on average) a larger number of somewhat likely clases, i.e., it assigns a dynamic number of classes to uncertain samples. PASLE optimizes entropy, we optimize (weighted) logits.  **We added a discussion on conceptual differences in the related work and also an empirical comparison, which shows that we outperform PASLE and others.**
>
> *The experimental section lacks comparisons with any baseline methods... especially from adjacent domains such as test-time adaptation.*
> > We treat any public pre-trained model as baseline that we aim to improve for any prediction of an in-domain sample.
> > We adjusted other methods such as [1] to our problem. We apologize for not doing so earlier, but related works, i.e., **Tent - ICLR 2021[2], even mentioned that their method makes no sense in our context:** "Note that optimizing a single prediction has a trivial solution: assign all probability to the most probable class." **We believed them. We see/saw our method more as complementary to domain adaptation methods, as we do not allow for continual learning**. That is, i.e., we use very high learning rates and reset the model after each sample.
>
> Ref [2] TENT: FULLY TEST-TIME ADAPTATION BY ENTROPY MINIMIZATION, ICLR 2021

---

> > ### Author Response · Authors · 2025-11-28
> >
> > Please tell us if anything remains unclear. The prior omission reflects the different problem scope also shared, e.g., by authors of [2]. We now include adapted baselines, where our method performs best, and kindly ask for reconsideration.

---

### Official Review · Reviewer_moFk · 2025-10-31

**Soundness:** 3
**Presentation:** 2
**Contribution:** 2
**Rating:** 4
**Confidence:** 4

**Summary:**

This paper studies whether focusing on likely classes can improve model predictions during test time. The authors introduce two simple fine-tuning strategies—iFo (increasing focus) and doFo (decreasing out-of-focus)—that apply one gradient descent step only when the model shows high uncertainty (measured by the top-1/top-2 probability gap). iFo aims to enhance shared features among likely classes, while doFo suppresses unlikely ones. Experiments across 70+ model-dataset pairs (vision and language) show that iFo consistently improves accuracy, while doFo often does not.

**Strengths:**

Novel yet simple idea – The “focus on likely classes” concept is intuitive and differs from classical entropy minimization or confidence-based TTA.

The method is tested on diverse image (CNNs/ViTs) and text (GPT-2, LLaMA-3, Gemma-3, etc.) models, showing broad empirical coverage.

Only one gradient step and per-sample adaptation make the method computationally efficient.

**Weaknesses:**

Although the paper cites Tent and related works, it does not directly compare with them in experiments (e.g., Tent, TTT++, CoTTA, etc.). As a test-time method, more quantitative comparisons with test-time adaptation approaches would strengthen claims.

The reported gains are relatively modest (e.g., +0.1–0.3%), which raises concerns about their practical significance. While the aggregated metrics (mean, standard deviation, and p-values) provide some support, they are not entirely convincing. In this context, comparisons with direct most-likely-class-based approaches such as Tent and ReCap [Region Confidence Proxy for Wild Test-Time Adaptation, ICML 2025] would be particularly important to better demonstrate the method’s advantage.

**Questions:**

How does the proposed method perform on out-of-distribution (OOD) datasets (e.g., ImageNet-Corruption), which are commonly used for evaluating test-time learning approaches?

More detailed and clearer ablation studies analyzing the effectiveness of iFo and doFo would also be preferred.

---

> ### Author Response · Authors · 2025-11-24
> **We solve a different problem (single in-domain sample); thus, comparison against TTA is non-obvious but now added**
>
> **Thanks**
> *It does not directly compare with cited methods in experiments (e.g., Tent, TTT++, CoTTA, etc.)*
> > - **Tent's authors stated in their paper that using Tent makes no sense for our problem:** "Note that optimizing a single prediction has a trivial solution: assign all probability to the most probable class." We agreed with them.
> > - We focus on a single in-domain sample without auxiliary data/tasks; the cited (and other methods) focus on domain adaptation using many samples (often with auxiliary data/tasks). **We saw our method more as complementary to the given ones and others**, as we are not focused on continual learning but on just fixing one potentially wrong uncertain prediction (due to our high learning rates, we reset the model after every sample). Still, **some like Tent can be tweaked to run for our problem. Results in the revised paper indicate that we outperform**. We also clarified differences to other works with a table and better described our problem in the revised version.
>
> *The reported gains are relatively modest.*
> > **Gains are expected to be small as we only optimize based on one sample**. One might say that it is surprising that there are gains at all. Others (like Tent's authors) did not think so. But **'gains per sample' look good**, e.g., a per sample a gain of 0.1% for a method using just 1 sample is an order of magnitude better than a method using 10,000 gaining 10%, i.e., 10%/10,000 = 0.01%.
>
> *Aggregated metrics are not convincing. Compare with other TTA.*
> > As stated, we added comparisons and we outperform. We also provide p-values for image and text data showing p-values for each below 0.01. This number is below common standards in science, which are often at 0.05. Also, our assumptions for testing are conservative, and the setups explicitly aim at counteracting potential overfitting, e.g., we use the same learning rate for all image models. Still, we expanded with additional tests strengthening our outcomes.
>
> *How does the proposed method perform on out-of-distribution (OOD) datasets (e.g., ImageNet-Corruption), which are commonly used for evaluating test-time learning approaches?*
> > Interesting question that we look into right now. Note that most techniques for OOD don't evaluate on in-domain samples, i.e., regular ImageNet, so we did not think of evaluating on out-of-domain samples as this is uncommon for techniques focusing on in-domain samples.
>
>
> **Please let us know, if things should be further clarified.**

---

### Official Review · Reviewer_utqt · 2025-11-01

**Soundness:** 3
**Presentation:** 2
**Contribution:** 2
**Rating:** 4
**Confidence:** 5

**Summary:**

This paper proposes two test-time fine-tuning methods, Increasing outputs for Focus classes (iFo) and Decreasing outputs of Out-of-Focus classes (doFo), to improve predictions on uncertain samples by focusing on likely classes via a single large-step gradient descent. An uncertainty assessment based on probability differences triggers optimization only when needed. Theoretical analysis highlights how iFo amplifies shared features among likely classes.

**Strengths:**

1. Efficient single-step optimization with large LR approximates multi-step results, minimizing computational overhead.
2. Comprehensive evaluation across diverse models (e.g., ViTs, ResNets, LLMs like GPT-2, Llama) and datasets (ImageNet, OpenWebText, etc.), demonstrating consistent gains for iFo (e.g., up to 2.2% on WideResNet).
3. Ablations on hyperparameters (LR, uncertainty threshold, iterations) and comparisons (e.g., input tuning) provide thorough insights.
4. Practical applicability: No auxiliary tasks or source data needed, works on pre-trained models.

**Weaknesses:**

1. Main concern: The method's primary motivation and approach seem to have appeared in prior TTA work [1] (Selective Label Enhancement Learning for Test-Time Adaptation, ICLR); authors need to further explain and strengthen the novelty and advantages of their method.

2. Images should be optimized for display and layout, preferably using vector graphics.

3. Some typos exist, e.g., line 266 has an extra ".".

**Questions:**

1. Can the method integrate with existing TTA techniques (e.g., entropy minimization) for further gains?

2. What architectural factors (e.g., transformers vs. CNNs) influence gains, and why?

---

> ### Author Response · Authors · 2025-11-24
> **PASLE[1] aims at domain adaptation (many out-of-domain) samples; we (single, in-domain); Method also differs**
>
> **We appreciate the review. Thanks.**
> *The method's primary motivation and approach seem to have appeared in prior TTA work [1];*
> > - **Motivation novelty**: **We don't consider domain adaptation using many samples like [1] but improving at test time on a single in-domain sample**. This problem is novel and not well-studied. Our **motivating assumption is that shared features of likely classes of a single sample are more to be trusted**. PASLE's is that one should not use a single label for an uncertain sample. We updated the Abstract/Intro and provided tables to clarify.
> > - **Novelty of approach**:
>   [1] (PASLE) splits samples into uncertain and certain to ensure good performance (high accuracy) on many samples. PASLE optimizes on all samples. We split to save on computation and optimize only on uncertain.
>   We use just two focus classes, which are very likely true. PASLE uses a dynamic number of potential classes per sample, i.e., all which might be true, which is typically more than 2. PASLE optimizes entropy which is common. We optimize (weighted) logits. **We added a discussion on conceptual differences (see e.g., new Table in related work) and also an empirical comparison. We outperform PASLE and other TTA** on our problem.
>
> *Can the method integrate with existing TTA techniques (e.g., entropy minimization) for further gains?*
> > This is an excellent question and our approach can be combined with others. Actually, **we see our method as complementary to existing methods like [1]**, as our process of improving on a single sample might make the model unusable due to high learning rates and, thus, requires resetting it. In contrast, existing TTA methods aim at continual learning, which adds to their methodological complexity. We did not combine them, but we modified them for our setup and compare against [1], Tent, and SAR. We outperform all.
>
> *What architectural factors (e.g., transformers vs. CNNs) influence gains, and why?*
> > We have provided some thoughts on architecture in the Appendix A.3 (on Experiment 2). In short, we could not identify any patterns (neither for model type nor model size etc.).
>
> [1] Selective Label Enhancement Learning for Test-Time Adaptation. ICLR 2025
>
> **If there is anything else to clarify, please let us know.**

---

### Official Review · Reviewer_qYqW · 2025-11-01

**Soundness:** 2
**Presentation:** 2
**Contribution:** 2
**Rating:** 4
**Confidence:** 3

**Summary:**

This paper proposes test-time fine-tuning to improve uncertain predictions by focusing on likely classes. When uncertainty is high (difference between top-2 probabilities less than 0.16), two methods are applied: (1) iFo increases outputs of focus classes by maximizing their weighted logits, and (2) doFo decreases outputs of unlikely classes by minimizing their average logits. Single gradient step with large learning rate modifies logits. Evaluated on 70+ model-dataset pairs (ImageNet with CNNs/ViTs, 6 LLMs on 6 text corpora), iFo shows consistent 1-2% improvements while doFo fails.

**Strengths:**

**Simple and Practical**: Elegantly simple - single gradient step on logits when uncertainty is high. Uncertainty measure (difference between top two probabilities) requires no calibration. Architecture-agnostic with easy implementation (code in Appendix B.2). Requires only one extra forward-backward pass.

**Broad Evaluation**: 70+ model-dataset pairs across vision (ImageNet on ResNet/DenseNet/EfficientNet/MobileNet/ViT) and language (GPT-2, Llama, QWEN, Fox-1, StableLM, Gemma on diverse corpora). Output spaces range 1K-100K+ classes. Honest reporting of doFo failures adds credibility.

**Weaknesses:**

**Limited Novelty**: Test-time gradient adaptation is established in TTA/domain adaptation. Main distinction (multiple likely classes vs. single class) is incremental. No comparison with existing TTA methods (Tent, TTT, MEMO) or calibration methods (temperature scaling, Platt scaling). Single-step optimization is a practical trick, not a conceptual advance.

 **Modest Gains Without Context**: Consistent 1-2% improvements but missing: (a) wall-clock time overhead measurements, (b) comparison with simple baselines (ensembles, calibration), (c) whether gains justify deployment complexity, (d) failure rate analysis - what percentage worsen? Figure 6 aggregates results without showing variance or per-sample effects.

**Questions:**

Please refer to the Weakness

---

> ### Author Response · Authors · 2025-11-24
> **Different problem setting (single in-domain sample); thus, comparison against existing TTA is non-obvious but now included**
>
> **Thank you for the review.**
> *Limited Novelty*
> > We apologize for not making this clearer in the paper.
> > - **Problem** novelty: **We don't consider domain adaptation** with many out-of-domain samples like Tent, TTT, MEMO, etc., but **we focus on improving on the prediction of a single in-domain sample using only that sample**. This problem is novel and not well-studied. The Abstract/Introduction and a comparison table make this clearer.
> > - **Rationalization** novelty: For our problem, our rationale for algorithms is based on the assumption that **shared features of likely classes of a single sample are more to be trusted**, which is also novel.
> > - **Method** novelty: We agree that using gradients for TTA has been well studied in recent years; however, recent methods use gradients, e.g., ICLR 2025 [1]. Focusing on weighted likely classes and **optimizing for logits instead of entropy** is novel. We provide a comparison table under related work. Also, we outperform existing methods.
>
> *No comparison with existing TTA methods (Tent, TTT, MEMO) or calibration methods (temperature scaling)*
> > - **The authors of Tent stated that Tent makes limited sense for our problem**: "Note that optimizing a single prediction has a trivial solution: assign all probability to the most probable class." **We trusted them** :-) .
> > - **All TTA methods treat a different problem. We see/saw our method as complementary**, i.e., run any domain adaptation technique to continuously update the model; then for each uncertain test sample: copy the model, run our method using the sample and reset the model. Our method uses high learning rates, which is incompatible with gradual updates of those methods and requires resetting models after each step.
> > - But **some methods like Tent (with some tweaking) can be applied to our problem. The revised paper shows results indicating that we outperform them.**
> > - For calibration: This is a good point, as we also mentioned in our paper that "potentially better calibration might further improve." Our calibration (using softmax) is standard and well-known. We considered it out of scope or potential future work to use other methods. This aligns with other papers in TTA, which also only used a single method (e.g., ICLR 2025 [1]).
>
> *Single-step optimization is a practical trick, not a conceptual advance*
> > In a classical setup (large batch sizes etc.), yes. But the **training dynamics could be quite different when using only a single sample**. In the paper, we discuss the **double growth effect**, saying that possibly multiple iterations with small learning rates can be more unstable than having one step with a large learning rate. This is opposed to common wisdom in deep learning, where instability arises from large learning rates. We see this "theoretical" insight as a conceptual advance, though we did not study it in depth empirically.
>
> *Modest Gains Without Context*
> > **From the problem setting, one can only expect modest gains**, i.e., our training data is just one sample. **From a data efficiency perspective our method is potentially very efficient**, i.e., gains computed per sample, gains are large, e.g., per sample a gain of 0.1% for a method using just 1 sample is an order of magnitude better than a method using 10,000 samples gaining 10%, i.e., 10%/10,000 = 0.01% < 0.1%.
>
> *(a) wall-clock time overhead measurements*
> > We added a discussion in the Appendix. In short, **we are faster** as we do not optimize samples that are classified with high certainty. If we use our idea for other methods (Tent, PASLE in ICLR 2025 [1]), which all require practically the same time.
>
> *(b) comparison with simple baselines (ensembles, calibration)*
> > Our baselines are strong public pre-trained models. (Model) ensembles are not used in any work we are aware of, including those of the reviewer.
>
> *(c) Deployment?*
> > We can say that our method is considerably **simpler** than a number of others as it requires **no extra needs** in terms of data/auxiliary tasks. The only new hyperparameter is the uncertainty threshold, which mostly helps to limit computation.
>
> *(d) Figure 6 aggregates results without showing variance or per-sample effects.*
> > Figure 6 (in the original submission) shows variances. Two lines are counts that have 0 variance. Also, Figures 7/8 (original submission) show individual configurations (model/dataset), but we do not look further at individual samples, as the reviewer correctly points out. This would be interesting, but this is not commonly done (also many related works don't do it).
>
> **Please let us know if we can clarify further.**
>
> Refs: [1] Selective Label Enhancement Learning for Test-Time Adaptation. ICLR 2025

---

### Author Response · Authors · 2025-11-29
**Summary for Area Chair**

**All 4 reviewers share one main concern: Comparison with out-of-domain adaptation techniques** that perform continual learning with many samples.
- **We added the requested comparison and outperform** supported by statistical testing. Aside from the empirical comparison, the conceptual comparison was improved and extended covering ICLR 2025. See e.g., tables in intro, related work and evaluation, other text colored in red.
- **The requested comparison is not obvious. We treat a different problem than these techniques: Improving on prediction using just one in-domain sample without any auxiliary tasks and data.** The authors of Tent stated that their approach leads to a trivial solution for a single (out-of-domain) sample, which also kept us from evaluating on it.
- One reviewer spoke of modest accuracy gains. However, gains per sample (as we use just 1 sample to improve) are actually large.


**Contributions:**
- **Novel problem setting** that is very practical: No extra needs, in-domain sample.
- **Novel understanding of what makes a feature reliable** backed by a theoretical discussion given just the output of one sample.
- **Extensive evaluation** showing gains: image and text; more than 70 dataset/model pairs.
- **Methodological contribution**: Different loss function, etc.

**Tent (ICLR 2021) and PASLE (ICLR 2025) were received very positively at ICLR. Before the rebuttal our work was seen mostly as borderline** (4,4,4,2). **The reviewers’ main concern stemming from a misunderstanding of our problem setting was addressed. We hope for a more positive assessment.**

---

### Meta-Review · Area_Chair_7G6R · 2025-12-26

**Summary:**

The paper explores test-time fine-tuning to improve predictions for single in-domain samples by focusing on likely classes. While the reviewers acknowledged the extensive evaluation across vision and language models, the core concerns focused on the limited technical novelty and the modest empirical gains. Most reviewers found the approach, using a single gradient step on logits, to be an incremental variation of existing Test-Time Adaptation (TTA) techniques. Furthermore, the significance of the 0.1 ~ 0.3% accuracy improvements was questioned regarding their practical impact versus the computational overhead.
Based on these considerations, I recommend Reject. We appreciate the authors’ efforts in the rebuttal and revision, and encourage them to further refine the paper in line with the reviewers’ feedback.

**Reviewer Concerns:**

The rebuttal and revision partially addressed reviewer concerns by clarifying the intended problem setting (single in-domain sample test-time adaptation) and by adding experimental comparisons to adapted TTA baselines. However, key concerns remain outstanding. The proposed approach is still perceived as incremental relative to existing test-time adaptation methods, the reported performance gains are modest and not sufficiently contextualized to justify the added test-time optimization complexity, and the narrow problem formulation limits the scope of evaluation, making it difficult to assess robustness and broader applicability (e.g., out-of-distribution or corruption-style analyses) beyond the specific in-domain setting considered.

**Reviewer Scores:**

This submission received initial scores of 2, 4, 4, and 4. Given the initial average and that several key concerns remain partially unresolved after the rebuttal, I recommend Reject.

---

### Decision · Program_Chairs · 2026-01-26

Reject